# Spatial communication systems across languages reflect universal action constraints

Kenny R. Coventry [1] ✉, Harmen B. Gudde [1,2,3], Holger Diessel [4], Jacqueline Collier[1], Pedro Guijarro-Fuentes [5], Mila Vulchanova [6], Valentin Vulchanov[6], Emanuela Todisco[5,7], Maria Reile [8], Merlijn Breunesse [9], Helen Plado [8,10], Juergen Bohnemeyer [11], Raed Bsili [12,13], Michela Caldano[1], Rositsa Dekova [14], Katharine Donelson[15], Diana Forker [16], Yesol Park[17], Lekhnath Sharma Pathak [18,19], David Peeters [20,21], Gabriella Pizzuto [22], Baris Serhan[23], Linda Apse [24], Florian Hesse [25], Linh Hoang[4], Phuong Hoang[4], Yoko Igari[1], Keerthana Kapiley [19], Tamar Haupt-Khutsishvili [16], Sara Kolding [26], Katri Priiki [27], Ieva Mačiukaitytė[28], Vaisnavi Mohite [19], Tiina Nahkola[8], Sum Yi Tsoi [1], Stefan Williams[6], Shunei Yasuda[1], Angelo Cangelosi [23], Jon Andoni Duñabeitia [29,30], Ramesh Kumar Mishra[19], Roberta Rocca[31,32], Jurģis Šķilters [24], Mikkel Wallentin [26,31,33], Eglė Žilinskaitė-Šinkūnienė [28] & Ozlem Durmaz Incel[34]

The extent to which languages share properties reflecting the non-linguistic constraints of the speakers who speak them is key to the debate regarding the relationship between language and cognition. A critical case is spatial communication, where it has been argued that semantic universals should exist, if anywhere. Here, using an experimental paradigm able to separate variation within a language from variation between languages, we tested the use of spatial demonstratives—the most fundamental and frequent spatial terms across languages. In $n = 874$ speakers across 29 languages, we show that speakers of all tested languages use spatial demonstratives as a function of being able to reach or act on an object being referred to. In some languages, the position of the addressee is also relevant in selecting between demonstrative forms. Commonalities and differences across languages in spatial communication can be understood in terms of universal constraints on action shaping spatial language and cognition.

Speakers of different (spoken) languages share the same perceptual apparatus, so one might expect that the world's 7,000 or so living languages[1] may have evolved communication systems that also share common properties[2,3]. Yet the idea that there are universals in communication systems has been challenged with studies documenting extensive cross-linguistic variation in domains closely yoked to perception, including colour naming[4–8] and spatial communication[9–11]. Spatial communication is an important test case; as Evans and Levinson note in 'The myth of language universals', "spatial cognition is fundamental to any animal, and therefore if Fodor is right anywhere [that languages directly encode the categories we think in], it should be here"[12] (p. 436). However, extensive cross-linguistic variation has been

documented in the spatial domain[9–11]. For example, languages vary in the terms they have mapping onto containment and support relations (equivalents of 'in' and 'on' in English), carving up containment and support in different, though systematic, ways[9]. To give a second example, most accounts of spatial cognition assume the priority of egocentric over non-egocentric spatial relations (for example, 'The pen is to the left of the stapler'—on the left side from the speaker's perspective[13,14]), but it has been shown that some languages, including Tseltal and Guugu Yimithirr, prioritize allocentric, geocentric or absolute relations even in tabletop space (for example, 'The pen is uphill/north of the stapler')[10,11]. Such variation has led to the views that there are fundamental differences in the semantic parameters languages use to describe space[12], and that substantial cross-linguistic variation exists in the context of more abstract domain-general constraints[15].

Here we consider arguably the most fundamental words in all languages—spatial demonstratives (for example, 'this', 'that', 'here' and 'there' in English). Demonstratives occur in all languages[16,17] and are among the earliest words to appear in a child's lexicon[18–20]. They are multimodal, more intimately linked to eye gaze and gesture than other spatial terms[21,22]. Joint attention (shared gaze) between speaker and addressee usually immediately precedes spatial demonstrative use[23–25], and pointing obligatorily accompanies demonstrative forms in some languages (for example, Goemai[26] and Yucatec[27]). Moreover, it has been claimed that demonstratives are among the earliest forms in language evolution[16,28], consistent with the view that language may have evolved from gesture[29]. Demonstratives are therefore prime candidates to examine possible universal constraints across languages.

Typological studies across large samples of languages have shown that around half of the world's languages have binary demonstrative systems (as in English), while 40% or so possess a three-way demonstrative system (for example, Spanish), with the remaining languages employing four or more demonstrative contrasts (for example, Navajo)[16,30]. Accounts of demonstratives vary both within and between languages. A prominent debate in the literature is whether (spatial) demonstrative use is primarily associated with an egocentric, body-oriented strategy to direct attention to objects or places in space[31–37], or whether demonstratives are social and interactive terms rather than terms for spatial reference[38–43]. While these approaches are not mutually exclusive[19], they have framed recent discussions of demonstrative systems between and within languages.

Binary systems in linguistic treatments of demonstratives are often assumed to contrast distance, with a proximal term used for an object (referent) near a speaker and a distal term for an object far from a speaker[31–37] (Fig. 1, Demonstrative System 1). In English, it has been shown that the use of 'this' and 'that' maps onto the more precise reachable/non-reachable distinction[34–36]. 'This' is used more when the specific hand pointing at an object is able to reach it and hence act on it[36], and extending reach using a tool leads to an extension of the use of 'this' from the end of the hand to the end of the tool[34]. So rather than the position of objects in Euclidean space per se determining demonstrative choice in English, it is the ability to act on an object that is important, associated with the perceptual distinction between peripersonal/reachable versus extrapersonal/non-reachable space (served by different brain systems and involving action selection)[44]. Contact or control of an object by the speaker has been proposed as a universal feature of all demonstrative systems (distinct from relative distance)[37].

In contrast to accounts of demonstratives prioritizing egocentric distance and/or object reachability/action, a range of other parameters have been identified as important for demonstrative choice in some languages (including languages with binary systems[35]), such as the relative positions of the speaker and addressee (for example, Japanese[37,38]), object visibility (for example, Sinhalese[45]), the attention or gaze of the speaker and addressee (for example, Turkish[46,47]), and the elevation of the object's location in the environment, such as uphill/downhill (for example, Jahai[48]). Among these, the relative

**Fig. 1 | Conceptual models of demonstrative choice as a function of object distance and addressee position (manipulated in the experiments).** The speaker (S) and addressee (A) sit side by side (left) or opposite, facing each other (right). The three conceptual regions (1, 2 and 3) represent the peripersonal (PPS) of the speaker (1), a medium distance (2) out of reach of both speaker and addressee (irrespective of the position of the addressee), and a region (3) far from the speaker and addressee when they are aligned (left) or within reach of the addressee but not the speaker when they face each other (right). Demonstrative Systems 1–3 represent hypothetical models of demonstrative use in a three-term language structured in terms of egocentric distance alone (Demonstrative System 1), territories of the speaker and addressee (Demonstrative System 2) or egocentric distance and shared space (Demonstrative System 3). D1, D2 and D3 represent distinct demonstrative forms.

positions of the interlocutors and their respective territories have been widely proposed as the most important constraints on demonstrative choice across languages. Around 25% of languages have person-centred demonstrative systems, recognizing the territories of the speaker and the addressee[49]. For example, some models of the Japanese demonstrative system propose that there is a term for the territory near the speaker, a term for the territory near the addressee and a term for far away from the speaker and addressee[37,38]. Space is thus mapped onto the territories of the speaker and/or addressee rather than being only speaker-based (egocentric) (Fig. 1, Demonstrative System 2). Moreover, analyses of demonstrative systems in some languages recognize shared space[39–43], prioritizing demonstratives as social and interactive terms over the importance of demonstratives as referring spatial terms. Jungbluth, for instance, has argued that the relative positions of the speaker and addressee (that is, the configuration of the conversational dyad) fundamentally change how demonstrative forms are used in Spanish[39] (but see ref. 47). When the speaker and addressee are aligned (sitting side by side; Fig. 1, Demonstrative System 3), Jungbluth argues that Spanish demonstratives operate as (relative) distance-based terms, but when the speaker and addressee are face to face, demonstratives encode locations within or outside of the conversational dyad (specifically, 'este' ('this') is used for the shared space between the speaker and addressee, and 'aquel' ('that') for any location outside of the shared space). Similar accounts recognizing shared space have been proposed for some other languages[40–43].

Accounts of demonstrative systems both within and between languages are gleaned using radically different methods[50]—from work with small numbers of linguistic informants in the field (usually in single digits)[37,40,42,51] to experimental studies in more controlled settings[34–36,47]. The former approach has the benefit of eliciting demonstrative use in a wide range of naturally occurring settings, but it risks spurious generalizations from small numbers of participants. For example, in the most systematic cross-linguistic analyses of demonstratives to date[51], the number of linguistic informants tested ranged from three to six participants per language, and no statistical analyses of the results were conducted. Whether one can generalize from demonstrative use on the basis of data from such small samples remains to be established and begs the question of the extent to which studies purporting to examine linguistic diversity need to be able to separate individual variation within a language from variation across languages.

Here we test the importance of egocentric distance, (propensity for) action, and the positions of the speaker and addressee across 29 diverse languages with varied demonstrative systems (Fig. 2 and Table 1).

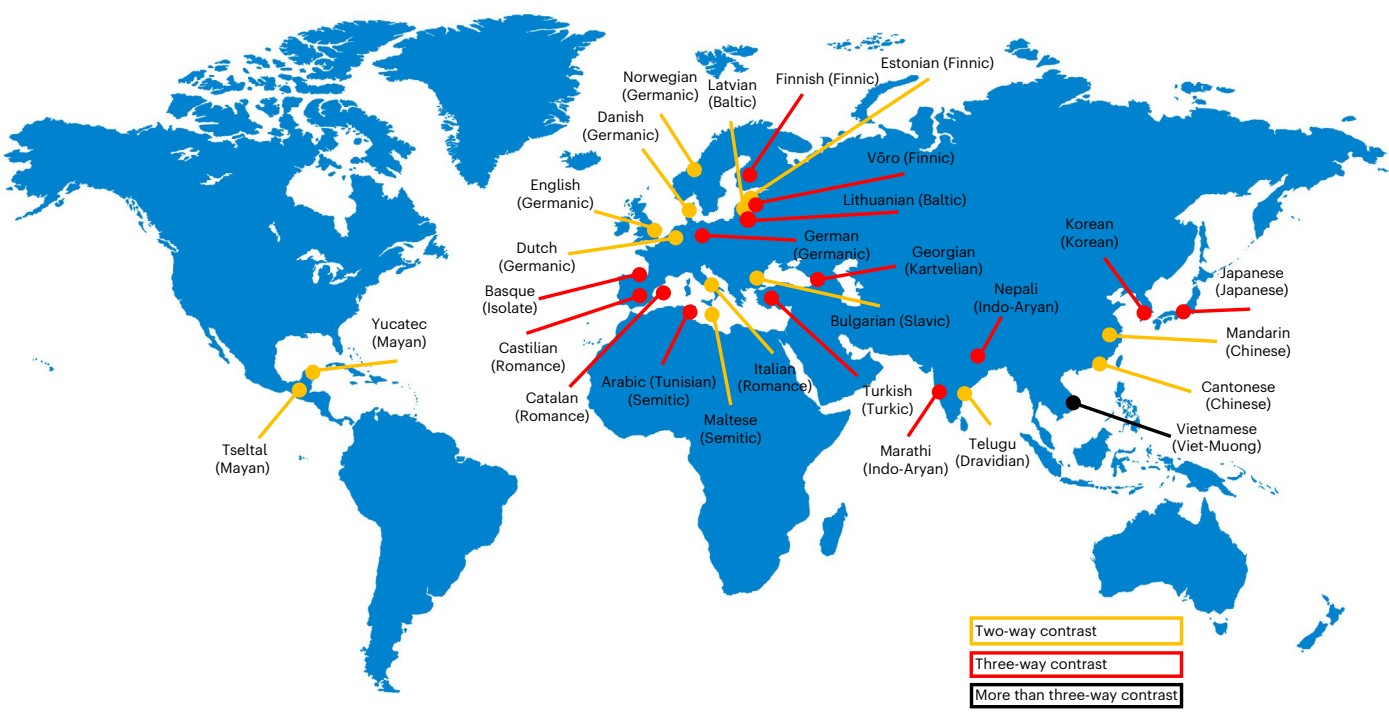

**Fig. 2 | Languages tested.** Language sample, classified by geography, language family and the number of demonstrative contrasts (see also Table 1). Credit for background map: rawpixel.com on Freepik.

The primary goal was to probe commonalities and differences across languages and to establish the extent to which demonstrative systems share any common constraints, and hence to ask whether any semantic universals can be identified for this essential linguistic category (that is, whether all languages have specific terms with common functions). Specifically, we tested the extent to which languages make proximal–distal distinctions mapping onto the (egocentric) distance a referent is placed from the speaker, versus sociocentric distinctions, as a function of the relative positions of the speaker and addressee. Unlike most previous studies examining the encoding of space across languages, often involving small numbers of linguistic informants that potentially limit generalizability, we adopted an experimental approach with sufficient power to separate individual variation from variation across languages. In doing so, a secondary goal was to identify the extent of inter-participant variability in demonstrative use within languages. We employed the 'memory game' method[34–36,52], in which participants were seated at a long table either beside or opposite an addressee (Fig. 1 and Supplementary Fig. 1). Down the midline of the table, coloured dots marked equidistant locations in front of the participants. On each trial, an object (a coloured geometric shape) was placed by the experimenter (the addressee) on one of the locations (see 'Materials'). The participants pointed at the objects and referred to them, choosing one of the demonstrative forms available in their language (for example, 'this/that black cross'; see Supplementary Information section 1 for the terms used for each language). The participants thought they were taking part in a study examining the effects of language on object-location memory, with memory probe trials maintaining the cover throughout the experiment, thus making memory the main focus of the study rather than demonstratives from the participants' point of view (see ref. 52 for more details).

The experiment manipulated the distance of the object from the participant (the speaker in the experiment) and the position of the addressee. Six of the marked locations on the table were used, creating three conceptual regions: Region 1, within the speaker's reach/peripersonal space (PPS), at 25 cm and 50 cm; Region 2, out of reach for both

the speaker and addressee (regardless of addressee position) and at medium distance from the speaker, at 150 cm and 175 cm; and Region 3, at 275 cm and 300 cm, furthest from the speaker but in the PPS of the addressee when the addressee was seated opposite the participant.

Languages were selected in accordance with four working criteria: (1) sampling across languages with demonstrative systems varying in the number of demonstrative terms, (2) sampling between and within language families, (3) sampling across geographical areas, and (4) the availability of researchers to collect data in the targeted languages. Data from 29 languages were collected (Fig. 2, Table 1 and Supplementary Table 1). The languages span geographical areas, genetic origins and differences in spatial communication systems. For example, they include languages that are typically considered to be person centred (such as Spanish and Japanese), languages that describe spatial relations using cardinal directions more than other languages (such as Tseltal) and languages with two, three or more spatial demonstrative terms.

## Results

The participants provided six responses for each of the region (three) × addressee position (two) combinations (collapsing across the locations within each region). Prior to data analyses, we removed demonstratives that were used infrequently (<2% of trials within a language), which in practice reduced the number of demonstrative forms in analyses to three for languages with more than three terms (Table 1). Analyses were performed using multinomial multilevel modelling, allowing us to partition the residual variance into a between-participant component and a within-participant component (the variance of the participant-level and response-level residuals, respectively) to accommodate clustered or grouped data[53–55]. All data and analysis scripts are available online.

The overall model (across all languages) tested the influence of distance and addressee position (fixed effects) in all our data, clustered around individual participants and individual language (random effects). The overall model produced a main effect of region ($F_{(4, 31,418)} = 2,476.749$, $P < 0.001$, $\eta_p^2 = 0.240$). The region effect coefficients show that both the distal and the third term are used more as

## Table 1 | Language sample

| Language | Region | Family | Genus | Number of terms |
|---|---|---|---|---|
| Danish | Denmark | Indo-European | Germanic | 2 |
| Dutch | Netherlands | Indo-European | Germanic | 2 |
| English | Britain | Indo-European | Germanic | 2 |
| German | Germany/Austria | Indo-European | Germanic | 3 |
| Norwegian | Norway | Indo-European | Germanic | 2 |
| Castilian | Spain | Indo-European | Romance | 3 |
| Catalan | Spain | Indo-European | Romance | 3 |
| Italian | Italy | Indo-European | Romance | 2 |
| Latvian | Latvia | Indo-European | Baltic | 2 |
| Lithuanian | Lithuania | Indo-European | Baltic | 4 |
| Bulgarian | Bulgaria | Indo-European | Slavic | 2 |
| Marathi | India | Indo-European | Indo-Aryan | 2 |
| Nepali | Nepal | Indo-European | Indo-Aryan | 2 |
| Estonian | Estonia | Uralic | Finnic | 2 |
| Finnish | Finland | Uralic | Finnic | 2 |
| Võro | South Estonia | Uralic | Finnic | 3 |
| Cantonese | China | Sino-Tibetan | Chinese | 2 |
| Mandarin | China | Sino-Tibetan | Chinese | 2 |
| Arabic | Tunisia | Afro-Asiatic | Semitic | 3 |
| Maltese | Malta | Afro-Asiatic | Semitic | 2 |
| Turkish | Turkey | Altaic | Turkic | 3 |
| Georgian | Georgia | Kartvelian | Kartvelian | 3 |
| Telugu | India | Dravidian | Dravidian | 2 |
| Basque | Spain/France | Isolate | Isolate | 3 |
| Korean | Korea | Altaic | Korean | 3 |
| Japanese | Japan | Altaic | Japanese | 3 |
| Vietnamese | Vietnam | Austroasiatic | Viet-Muong | 5 |
| Tseltal | Mexico | Mayan | Mayan | 2 |
| Yucatec | Mexico/Belize | Mayan | Mayan | 2 |

## Table 2 | Classification table for the generalized linear mixed model (multilevel model) including all languages (overall percentage correct, 82.2%)

| | | Predicted | | |
|---|---|---|---|---|
| | | **Proximal** | **Distal** | **Third term** |
| Observed | Proximal | 9,692 | 2,301 | 151 |
| | | 79.81% | 18.95% | 1.24% |
| | Distal | 1,174 | 14,928 | 483 |
| | | 7.08% | 90.01% | 2.91% |
| | Third term | 289 | 1,209 | 1,203 |
| | | 10.70% | 44.76% | 44.54% |

the object is placed further away (Tables 2 and 3 and Fig. 3). There was also a (small) overall effect of addressee position ($F(2, 31,418) = 6.277$, $P = 0.002$, $\eta_p^2 < 0.001$). To explore this effect further, we ran additional analyses using McNemar tests for change to test how the proximal demonstrative across languages was affected by region and addressee position (using Durkalski's method to adjust for the clustered nature of the data[56]). In this analysis, we therefore compared the use of

proximal demonstratives with that of non-proximal demonstratives (distal in two-term languages and both distal and the third term in three-term languages). There were addressee effects in Regions 2 and 3 ($\chi^2 (N = 5,443) = 28.809$, $P < 0.001$; $\chi^2 (N = 5,443) = 9.166$, $P = 0.002$, respectively), where there is a slight tendency overall for the speaker to switch from the distal or third term to the proximal term when the addressee is opposite the participant. There was no addressee effect in Region 1 ($\chi^2 (N = 10,888) = 2.132$, $P = 0.144$). The region × addressee position interaction was not significant ($F(4, 31,418) = 0.945$, $P = 0.437$, $\eta_p^2 < 0.001$).

Importantly, both random effects were also significant (Supplementary Tables 3 and 4): there was significant variability in demonstrative use between languages (with significant effects for both the relation of the proximal demonstrative with the distal and that of the proximal with the third term; Supplementary Table 3).

To explore the between-language variability, we further tested the languages in individual models. Demonstrative use for each individual language was analysed using multinomial multilevel modelling (which is binomial in two-term languages) of the region × addressee position combinations. Satterthwaite approximation was used to control for differences in sample variances[57], and classification tables were used to assess overall model accuracy. A priori, all interactions were entered into the model, and to keep models comparable between languages, non-significant interactions were retained in the model. The main predictors (addressee position and distance of the object from the speaker) were entered into the multilevel model with data structured by participant, ensuring that within-participant effects were accounted for in the model (random effects). The reference categories for the models in the main analyses were the proximal demonstrative (for example, 'this'; for each language, we checked whether the assumed proximal demonstrative was indeed the most frequently used demonstrative for the PPS region), the closest region (in the speaker's PPS) and addressee position as side by side. For all languages, we tested main effects (of distance and addressee position) and then whether effects occurred when we compared Region 1 versus Region 2, or Region 1 versus Region 3, and the interactions.

For all three-term languages, we also tested whether the effects occurred for either the distal or the third term (both compared to the proximal term). The third demonstrative may serve different purposes between different languages (for example, typological analyses suggest that some may function as a medial term, some as a perspective-taking term and so on). The actual pattern of use for this third term must be determined in an individual analysis.

Figure 4 shows the pattern of factors influencing demonstrative choice by language. The final models revealed a main effect of distance in all languages (all $P < 0.001$ for the complete models). All tested languages have a proximal term used significantly more in Region 1 than in Regions 2 and 3 (both $P < 0.001$ for all (complete model) languages) and a distal term that is used most frequently in Regions 2 and 3 (Fig. 3c). Moreover, across all languages, the proximal term shows a steep drop-off from Region 1 to Region 2, with an average of 74% of all proximal use in Region 1 versus 18% in Region 2 and 8% in Region 3. While the results are consistent with a mere distance effect, the steep drop-off in proximal term use when the referent is outside of reach suggests that the use of proximal and distal terms across all languages specifically maps onto a reachable/non-reachable (peripersonal/extrapersonal space) distinction, consistent with previous results from English and Spanish[34–36] (which manipulated reachability directly). Moreover, the fall-off in the use of the proximal term across languages closely matches data from non-linguistic tasks, suggesting a parity between non-linguistic processing of space and demonstrative use to talk about object location (see Supplementary Information section 4.1.1 for further discussion).

In addition to a main effect of distance in every language, in eight languages (five three-term languages and three two-term languages), there were main effects of addressee position and/or

**Table 3 | Fixed coefficients for the overall model**

| | Coefficient | s.e. | t | 95% CI | Exp(coefficient) | 95% CI for Exp(coefficient) |
|---|---|---|---|---|---|---|
| Intercept: distal | −2.319*** | 0.246 | −9.433 | (−2.813, −1.825) | 0 | (0.06, 0.161) |
| Position opposite | −0.06 | 0.065 | −0.937 | (−0.187, 0.066) | 0.349 | (0.829, 1.068) |
| Region | | | | | | |
| Region 3 | 4.94*** | 0.074 | 66.443 | (4.794, 5.086) | 0 | (120.796, 161.669) |
| Region 2 | 3.684*** | 0.063 | 58.769 | (3.561, 3.806) | 0 | (35.188, 44.989) |
| Interactions | | | | | | |
| Opposite×Region 3 | −0.089 | 0.098 | −0.904 | (−0.282, 0.104) | 0.366 | (0.754, 1.109) |
| Opposite×Region 2 | −0.127 | 0.084 | −1.508 | (−0.291, 0.038) | 0.131 | (0.747, 1.039) |
| Intercept: third term | −7.034*** | 0.768 | −9.157 | (−8.606, −5.463) | 0 | (0.000, 0.004) |
| Position opposite | 0.014 | 0.137 | 0.099 | (−0.254, 0.281) | 0.921 | (0.776, 1.325) |
| Region | | | | | | |
| Region 3 | 4.372*** | 0.126 | 34.628 | (4.124, 4.619) | 0 | (61.827, 101.419) |
| Region 2 | 3.934*** | 0.117 | 33.605 | (3.704, 4.163) | 0 | (40.622, 64.277) |
| Interactions | | | | | | |
| Opposite×Region 3 | −0.07 | 0.17 | −0.416 | (−0.403, 0.262) | 0.678 | (0.668, 1.299) |
| Opposite×Region 2 | −0.197 | 0.158 | −1.248 | (−0.508, 0.113) | 0.212 | (0.602, 1.119) |

The reference categories are as follows: demonstrative, proximal term; position of addressee, side by side; region, Region 1. ***$P<0.001$. CI, confidence interval. Exp, exponential.

position-by-distance interactions, albeit with small or very small effect sizes[58] (Figs. 4 and 5). For five of the three-term languages (Finnish, Georgian, Japanese, Korean and Lithuanian), addressee effects were present in Region 3, with one term (the distal or third term) used more when the addressee was opposite the speaker, and a corresponding drop in the use of the other term (Supplementary Table 2). In this region, when the addressee is seated opposite the speaker, the referent is reachable by the addressee but not by the speaker. The additional distinctions made by speakers of these languages thus extend the concept of reachability/non-reachability to a dyadic partner[59]. This supports the view that some three-term languages have medial/distance terms while others do indeed have a specific term consistent with person-centredness[60]. In contrast to the three-term languages exhibiting addressee effects, the three two-term languages (Bulgarian, Italian and Mandarin) show addressee effects in Region 2, with the proximal term used more when the addressee is opposite the speaker than when the speaker and addressee are side by side. Thus, addressee effects for two-term languages are strongest when the referent is out of reach of both the speaker and the hearer, but in shared space.

Finally, there was significant inter-participant variability for every language tested (Supplementary Table 4). This is important as it tells us that within each language, participants differed significantly in how they used distal terms (and third terms, if applicable) relative to proximal terms. So if one were to test only a small number of participants, as is typical in anthropological studies, making generalizations about demonstrative use in speakers of the language as a whole would be spurious for this semantic category.

## Discussion

Spatial demonstratives occur in all languages, and it has been argued that they are among the oldest recorded words in language evolution. Yet challenges to the view that spatial communication systems across languages share universal constraints have not previously considered demonstratives using methods and sample sizes powerful enough to detect semantic universals should they be present.

Our primary goal was to test the extent to which languages make proximal–distal distinctions mapping onto the (egocentric) distance a reference is placed from the speaker, versus sociocentric distinctions as a function of the relative positions of the speaker and addressee.

The results reveal striking similarities (as well as some differences) across languages in how available demonstrative terms within languages map onto space. All tested languages have a demonstrative form that maps onto peripersonal/reachable space and a second form that is used for extrapersonal/non-reachable space. The data are thus consistent with the discovery of a semantic universal for this essential linguistic category, with all languages expressing the distinction between reachable and non-reachable space[34–37]. The importance of the reachability versus non-reachability of an object as a universal constraint on spatial communication systems is in line with approaches to the origins of language and theory of mind through action. It has been proposed that a child moves from reaching/not reaching to pointing behaviour and ultimately to symbolic communication from gestures (ontogenetic ritualization[61,62]). The links between reaching and demonstrative use across languages bolster the view that action, pointing and language are interrelated and are core to the use and development of language universally. Demonstratives are the terms in language that most closely map onto this process, and their universal properties point in the direction of support for gesture as central to both language learning and language evolution. As languages evolve, one might expect that this basic action component remains intact while other distinctions may come and go. There is some evidence to support this in a study on shrinking in the demonstrative system of Spanish–Norwegian bilinguals who maintain the peripersonal/extrapersonal space contrast over time but lose the use of the third distal term in their native (Spanish) language[63]. This suggests that languages with many demonstrative contrasts may reduce over time while nevertheless preserving the basic action system contrasts underlying space.

The presence of a common egocentric proximal/reachable–distal/non-reachable distinction across languages reinstates the centrality of egocentric spatial relations in spatial language and spatial cognition. The findings that some languages, including Tseltal and Guugu Yimithirr, prioritize allocentric, geocentric or absolute relations even in tabletop space (for example, 'The pen is uphill/north of the stapler')[10,11] have been extremely influential in the argument for diversity of languages and spatial cognition across cultures, challenging the primacy of egocentric space. Yet such findings have been gleaned from studies examining spatial adpositions across languages and have not considered spatial demonstratives. The early appearance of spatial

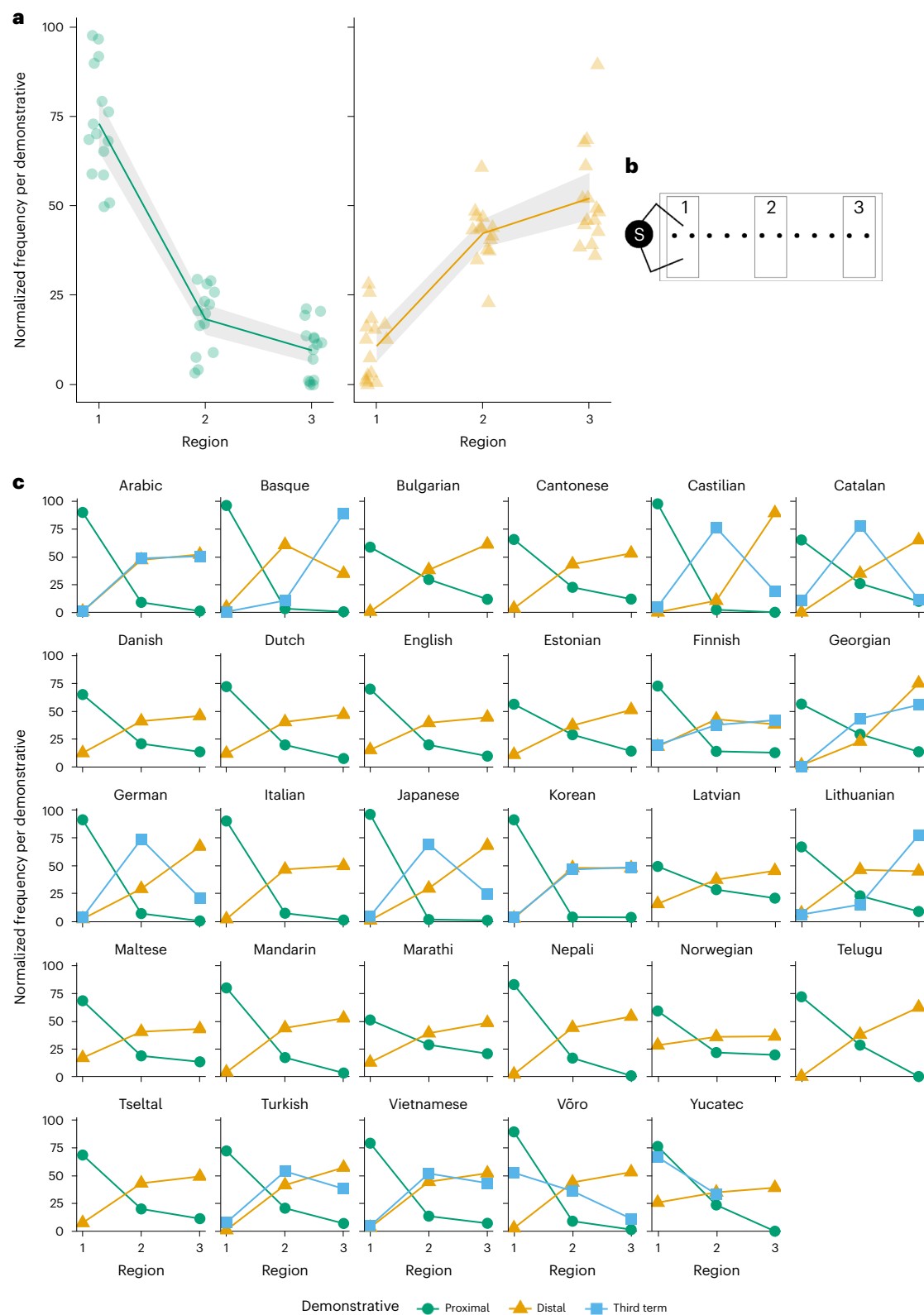

**Fig. 3 | Normalized frequency distributions of proximal and distal demonstrative forms in the overall analyses and for individual languages.** **a**, Normalized frequency distributions of proximal and distal demonstrative forms in the overall analyses. The shaded areas represent scaled confidence intervals. **b**, Key to the regions. **c**, Normalized frequency distributions of proximal and distal demonstrative forms for individual languages.

demonstratives in language development relative to other spatial terms[20] reinforces the centrality of egocentric space across languages.

We also find that some languages make distinctions in addition to the egocentric reachable/non-reachable distinction. Speakers of

some three-plus-term languages take addressee position into account when choosing demonstrative terms. In these languages, the effects of addressee position were most pronounced in the region where the object was reachable by the addressee, suggesting an extension of the

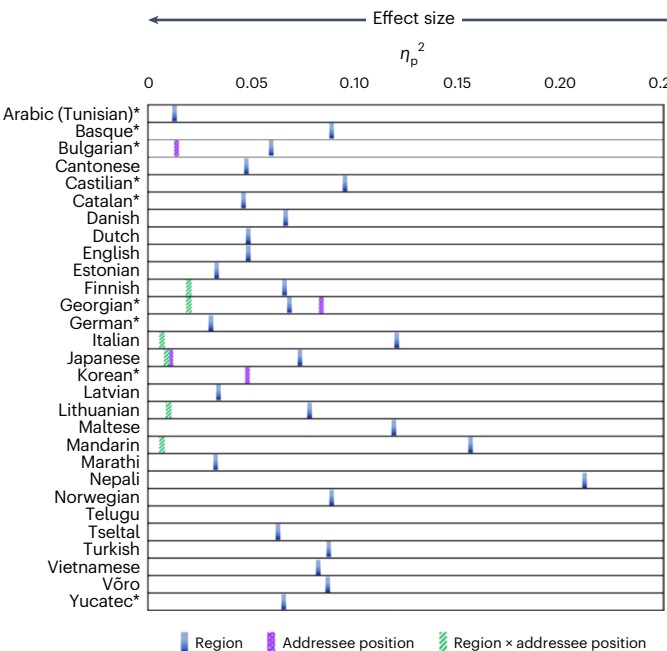

Fig. 4 | **Overview of statistical results across individual languages with effect sizes.** The effect sizes can be interpreted as follows: 0.01 is a small effect size, 0.06 is a medium effect size and >0.14 is a large effect size[58]. Asterisks (*) indicate follow-up models. Note that the final model did not include Region 1 in Korean; hence, the distance effect was absent in the final model though a distance effect is clearly present. Also note that no model would run for Telugu due to a dominant effect of distance (Supplementary Table 2).

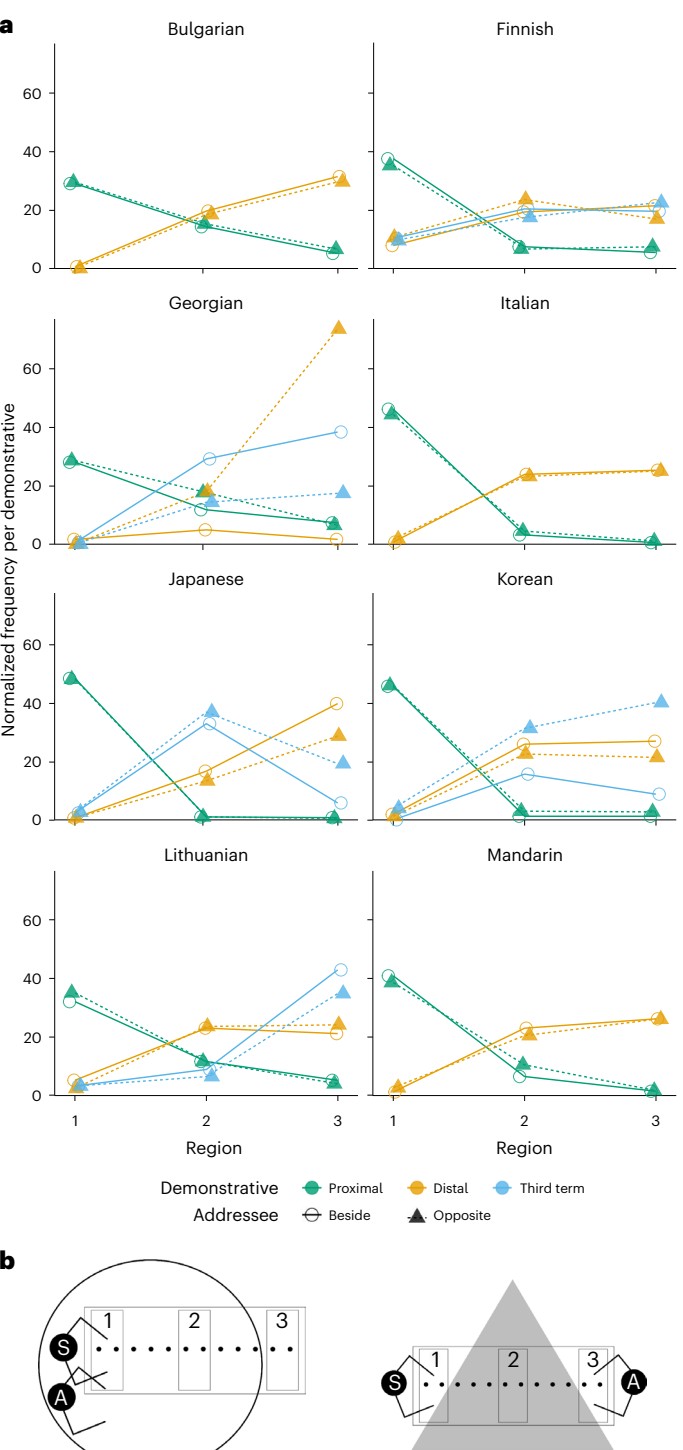

Fig. 5 | **Normalized frequency distributions of demonstrative forms for individual languages with addressee position effects and/or interactions between addressee position and region. a**, Normalized frequency distributions. **b**, Key to the regions.

reachable/non-reachable distinction to a conspecific. Moreover, while some languages have specific terms to mark location from a conspecific's perspective, speakers of languages that do not may nevertheless be sensitive to addressee position. In addition to addressee effects in a small number of two-term languages, there was a tendency across all languages for the proximal term to be used more when the addressee was seated opposite the speaker and the object was out of reach of the speaker than when the object was out of reach and the speaker and addressee were aligned. The existence of some languages with a specific term to mark perspective sits alongside the potential use of perspective taking in languages lacking specific terms to do so.

The secondary goal was to identify whether there is inter-participant variability in demonstrative use within languages. Practically, it is difficult to recruit samples of participants in many languages in the field, and it is therefore understandable that many previous studies have involved low numbers of linguistic informants and accordingly fail to consider diversity in demonstrative use within a language. However, all 29 languages in our analyses showed significant inter-participant variability in how demonstratives are used within languages. Spatial demonstrative use seems to be flexible, setting demonstratives apart from other spatial terms, including spatial adpositions and motion verbs, where there is agreement between participants regarding how 'in', 'on' and 'over', for example, map onto space. Differences in past approaches to demonstratives both within and between languages may be at least in part a result of variation within languages. We echo past concerns[64–66] that linguistic theory based on linguistic intuitions from small numbers of informants is on shaky ground; identifying generalizations within and between languages requires the application of statistical methods that can separate variability within a language from variation between languages.

Perspective taking within languages may also be one of the main reasons for significant variability within individual languages. In deixis, switching between the addressee's body as the deictic centre and one's own body (egocentric perspective)[67] is likely to be affected by individual preference as well as the degree to which the speaker and addressee are involved in a joint task[59]. Demonstratives may be regarded as similar to other spatial terms—the so-called 'projective' adpositions (for example, 'to the left/right' and 'in front of')—where it has been shown that another person's perspective (for example, the addressee's left, not the speaker's left) is often used to assign direction when the speaker

and addressee are misaligned[68,69]. For example, when describing the position of an object relative to a person in a photograph, around 25% of English-speaking participants chose to describe the location from the perspective of the person in the photograph ('on the left' (of the person in the picture, rather than the participants' left)[68]). Hence, there is substantial variation in the choice of reference frame adopted for spatial language, and our results suggest that demonstrative use is similar. Another possible origin of inter-participant variability may also relate to what has been termed 'neutral'[22] or 'default'[20] demonstrative use. In English, for instance, 'that' can be used at any object location (hence, it can be used as a neutral distance marker), and accordingly 'this' is not always used at proximal locations to the speaker. It is unclear to what extent individual languages have such neutral/default terms and how frequently they are used, but this is likely to be a source of both inter-participant variability and interlanguage variability.

The pattern of constrained variation for spatial terms across languages is broadly consistent with variation of other linguistic categories closely yoked to perception[15]. However, what sets demonstratives apart from other linguistic perceptual categories, such as colour, is that demonstratives do not just refer to abstract features of the perceptual world but make distinctions that prioritize agency, and specifically how agents are able to act on objects in the world, in addition to contrasting geometric regions of space. While the communicative need for other categories may vary across cultures, action is an arena where speakers across languages do not differ; hence, variation across languages is lower.

Future work would do well to consider other variables that have been argued to affect demonstrative choice in some languages, including features of the environment (for example, elevation[48]), object properties such as visibility[35,45], ownership[16,35], familiarity[35], and the gaze directions of the speaker and addressee[46,47], to assess whether such variables affect demonstrative choice in only a subset of the world's languages or whether indeed there is constrained variation. For example, while joint attention is often regarded as a precondition for demonstrative use[23–25], some have argued that demonstratives have the role of reorienting attention (such as when the addressee is not attentionally engaged)[46,47,70]. In the present study, joint attention was consistent throughout, and hence the possible universal role of demonstratives vis-à-vis attention manipulation was not considered. How demonstrative use changes as a function of disengagement of the addressee could be easily tested with modifications to the memory game paradigm, allowing the exploration of attentional interaction changes between the speaker and addressee. It is also important to note that demonstratives (in common with other types of spatial expressions) are also used in non-spatial domains, including time (for example, 'this/that day'), and discourse deixis (where written language unfolds in time, overlapping with temporal deixis). Talmy[71] has argued that the use of demonstratives in discourse involves the same cognitive processes as the perceptual use of demonstratives. However, commonalities and differences in discourse deixis across languages have not yet been examined.

Finally, it is important to situate the present work in the context of the range of methodologies that one can use to explore the use of words both within and between languages. The use of a strictly controlled experimental method testing large numbers of participants does come at a cost, in that the sample of languages tested is not as diverse as one might ideally like, and the experimental setting itself is necessarily constrained. In contrast, field workers can sample language use across a wide range of spatial and conversational settings, at the cost of loss of statistical power and control. These are not competing methods, however: the rich insights from linguists working in the field can be regarded as generating testable hypotheses for more high-powered and controlled studies, and as such the methods are complementary.

In summary, challenges to the view that spatial communication systems across languages share universal constraints have not previously considered demonstratives and have failed to consider the role of action in spatial communication. All the languages we tested have a demonstrative form that maps onto (egocentric) peripersonal/reachable space and a second form that is used for extrapersonal/non-reachable space. Space ultimately serves action, and it is this action component that serves as a bridge between physical and conceptual models of space. This essential action component for demonstratives may link to their early evolutionary origin as linguistic forms, with action as a potential driver for the development of linguistic systems.

## Methods

Prior to data collection, the study received full ethical clearance from the University of East Anglia's School of Psychology Ethics Committee (approval numbers 13-14-5 and 2017-0034-000748, granted on 9 March 2015 and 8 September 2017, respectively) covering data collection across languages. Local clearance was also required for Finnish data collection (from Tartu University, approval number 293/T-21, granted on 20 May 2019). All procedures were carried out in accordance with the guidelines of the British Psychological Society, American Psychological Association, Association for Psychological Science and the Declaration of Helsinki. All participants provided either written or verbal (recorded) consent prior to testing. The data were collected between January 2016 and December 2019, with staggered testing of languages during that period (so all testing sites used the same apparatus).

### Sample

Data were collected from 29 languages, spanning geographical areas, genetic origins and differences in spatial communication systems (Fig. 2 and Table 1). A statistical power analysis (a priori) was performed for sample size estimation using G*Power (version 3.1) (ref. 72). With power = 0.9, $\alpha$ = 0.05 and the effect sizes reported in Coventry et al.[35], the projected sample size is approximately $N$ = 17 for each language. Given that the effect size was based on English and that many languages tested have no empirical data on demonstrative production, we set 17 participants as the minimum sample size while aiming for 30+ per language ($N$ = 914, mean = 32 participants per language). A total of 914 participants took part, and 874 participants were included in the analyses (487 female (self-reports); mean age, 26; s.d., 7.64; Supplementary Table 1). The participants took part for nominal payment, for course credit or on a voluntary basis (commensurate with cultural norms of participation for each language). The data from 40 participants were excluded from the analyses on the basis of a priori criteria for exclusion as follows: (1) participants did not have normal or corrected-to-normal vision, (2) participants guessed that the study was about demonstrative use, or (3) participants reported deliberately using demonstratives in the study in a way they would not normally use them.

All researchers participating in data collection were trained linguists, psycholinguists or cognitive scientists with expertise in the specific language being tested. They were responsible for participant recruitment and data collection at each site, sometimes working with an additional researcher to collect the data (fully trained to do so). All communication with participants was in the first/native language of the participants.

### Materials

The experiment employed the 'memory game' method[34–36,52]. This established method was designed to elicit language under strictly controlled experimental conditions, but without participants being aware that language data were being collected. To do so, the participants were told that they were taking part in a study on the effects of language on object-location memory, with memory probe trials maintaining the cover of the memory experiment throughout. Such a method makes memory the focus of the study rather than demonstratives from the participants' point of view (see ref. 52 for more details).

The participants were seated at a large table (325 cm long), with 12 marked locations (coloured dots) spaced equidistantly 25 cm apart

down the midline of the table directly in front of the participants, starting at 25 cm from the participants' edge of the table (Supplementary Fig. 1). On each trial, the experimenter (the addressee) placed an object on one of the coloured dots. The objects placed were coloured shapes on disks. All shapes were basic geometric shapes: a black cross, a green star, a yellow triangle, a red circle, a blue heart, an orange square, a red sun, a white moon, a red moon and a black bar. In each language, six objects from the set were selected (four for Tseltal, given the available colour terms), ensuring that the language had a colour lexicon able to differentiate the colours, and ensuring that all objects were matched for gender (in gendered languages).

The experiment manipulated the distance of the object from the participant (the speaker in the experiment) and the position of the addressee. The addressee was seated either next to or opposite the speaker. This addressee position manipulation was blocked and counterbalanced: the addressee switched their position once, halfway through the experiment. The distance condition was pseudorandomized to ensure that no object or distance was used in two successive trials, preventing carry-over effects. Six of the marked locations on the table were used, creating three conceptual regions: Region 1, within the speaker's reach/PPS, at 25 cm and 50 cm; Region 2, out of reach for both the speaker and the addressee (regardless of addressee position) and at medium distance from the speaker, at 150 cm and 175 cm; and Region 3, at 275 cm and 300 cm, furthest from the speaker but in the PPS of the addressee when the addressee was seated opposite the participant.

Due to specific lab and field conditions, the dimensions of the table varied slightly across languages, but critically the described reachability of the three categorical regions was controlled in all languages: distances closest to the speaker and addressee were within their respective reach, the distances at the middle of the table were out of reach for both and the distance furthest from the speaker was equal to the distance furthest from the addressee. For example, the smallest table was used at the Nepali field site, where 9 locations rather than 12 were marked, meaning that the middle locations were respectively 100 cm and 150 cm from the interlocutors and thus still out of reach. Furthermore, Yucatec-speaking participants stood at the table (rather than being seated), as the room used was not large enough for the participants to sit.

## Procedure

The participants were told that the study was testing the effects of language on memory and that they were in a 'language' condition. Specifically, they were asked to point at and name the object placed on the table, using three words: [demonstrative] [object colour] [object shape name]—for example, "this black cross" (in the case of English). To ensure that the memory game cover was maintained, the participants were asked to remember the most recent location of each of the six objects throughout the experiment, and their memory during the experiment was tested (three blocks of four memory probes each were evenly spaced throughout the experiment).

In the instructions given, the participants were told that it was important to use the same amount of language on each trial, so that it was the same for everyone taking part. On each given trial, they were instructed to use one of the available demonstratives within their language—for example, either 'this' or 'that' plus colour plus shape (such as "this/that red square") for English. The instructions never referred to the use of a specific demonstrative at a specific location to ensure that the participants were not primed to use demonstratives in specific ways. The participants completed six practice trials (one trial for each of the six locations, but in pseudorandomized order) to get used to the task, and in particular to help maintain the memory cover (a memory probe was given at the end of the six practice trials—for example, "What was the most recent location of the black cross?"). The participants were not given any feedback at the end of the practice trials regarding how they used demonstratives, except that they should remember to use all of the available demonstratives during the task (if they had not already done so).

Prior to data collection for any of the languages, all researchers were trained in the method to ensure equivalence of testing across languages. The primary vehicle for training was a video publication describing the background and providing instructions on how to run the experiment[52], supplemented with Skype and email briefings and clarifications as required. In addition, researchers within the EU H2020 ITN DCOMM framework attended a training session on the memory game procedure.

The researchers were provided with a transcript of the instructions, trial lists and debrief questions. The lists comprised 36 trials (addressee position (two) × distance (six) × three repetitions per cell of the design), ensuring that no object or location was the same in any directly successive trial. At debrief, all participants were asked whether they understood what the study was about to ascertain whether they had figured out the manipulations, to determine whether they had a strategy for their demonstrative use and to establish whether they could reach (only) the first two locations. If they showed knowledge of the study that could lead to demand characteristics or could not reach (only) the first two locations, the protocol required that their data were excluded. The researchers translated the provided materials into the target language. From the moment the participants entered the testing room, all communication was in the first/native language of the participants.

All sites were instructed to pilot the study with participants to ensure fine-tuning of the instructions in the specific language and to become familiar with the testing protocol. In particular, we wanted to ensure that participants across sites would not realize that language data were being collected, and hence we checked that all sites included memory probe trials and debrief questions tapping whether the participants were aware of the collection of language data (those participants were eliminated from the analyses; Supplementary Table 1). Another key check was that participants across sites were instructed to use all demonstrative forms available to them in their language; they were prompted twice during instruction and again after the first memory probe (if required).

## Statistical methods

The data analyses were performed in SPSS version 27 (ref. 73). To accommodate differences in variance between groups in the individual language analysis, we employed Satterthwaite approximation to calculate the effective degrees of freedom[57]. This allowed the model for each language to account for low numbers in some response categories. However, while binomial and multinomial multilevel models are the appropriate tests for modelling clustered data with categorical outcomes, it is also important to be aware that where the data are such that perfect prediction of an outcome occurs from one or more covariates, a feature known as separation occurs[74]. If one or more groups of the outcome variable are perfectly separated by the predictor(s) (that is, they never or always occur), then unrealistic coefficients will be estimated, and effect sizes will be greatly exaggerated[75]. Cook et al.[74] demonstrate that the consequences of failing to recognize such separation in multinomial and binomial models include implausible parameter estimates such as the relative risk ratio being in the hundreds of thousands. They highlight that separation can be identified from simple cross-tabulation of the data, where, for one or more combinations of the independent variables, some category of the dependent variable will occur with frequency zero—meaning there is no variation in that category.

As can be seen from the cross-tabulations in Supplementary Table 2, in some of the languages, zeros are present for a demonstrative under one or more combinations of the independent variables. Hence, the binomial or multinomial multilevel models for these languages produced implausible parameter estimates, and the planned analysis should be considered as producing unreliable coefficients.

We therefore followed up these specific models with a posteriori models excluding the individual region or demonstrative with zero values. These languages are marked with asterisks. For transparency, the a priori models are also represented in Supplementary Information section 4.3. Supplementary Information section 4.2 presents the analysis for each individual language.

## Reporting summary

Further information on research design is available in the Nature Portfolio Reporting Summary linked to this article.

## Data availability

All data are available at the following link: https://osf.io/ush2w/?view_only=1f38fa7ae6ce4bbab456eee80615ebe4.

## Code availability

All analysis code is available at the following link: https://osf.io/ush2w/?view_only=1f38fa7ae6ce4bbab456eee80615ebe4.

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

## Acknowledgements

We thank N. Bullet, P. Hartzler and A. Pathak for their help in recruiting participants; B. T. Sek, J. S. Ladislao, I. L. Hansen and A. S. Girón for assisting with data collection; and H. Mitterer and H. Khairi for facilitating data collection. N. Kishita provided input and translations for the Japanese data collection. This research was supported by EU H2020 ITN Marie Skłodowska-Curie Action grant agreement no. 676063 (DCOMM) awarded to K.R.C. and colleagues. The funders had no role in study design, data collection and analysis, decision to publish or preparation of the manuscript.

## Author contributions

Conceived and designed the project: K.R.C., H.B.G. and H.D. Drafted the article: K.R.C. Contributed to language selection: K.R.C., H.B.G., H.D., P.G.-F., V.V. and M.V. Analysed and interpreted the data: K.R.C., H.B.G. and J.C. Processed the data: H.B.G. and J.C. Organized and collected the data: H.B.G., E.T., M.R., M.B., H.P., J.B., R.B., M.C., R.D., K.D., D.F., Y.P., L.S.P., D.P., G.P., B.S., L.A., F.H., L.H., P.H., Y.I., K.K., S.K., T.H.-K., K.P., I.M., V.M., T.N., S.Y.T., S.W., S.Y., P.G.-F., V.V., M.V., A.C., J.A.D., R.K.M., R.R., J.S., M.W., E.Ž.-Š. and O.D.I. Provided detailed feedback on iterative drafts of the manuscript: H.B.G., H.D., J.C., J.B., M.B., M.C., D.P., G.P., M.R., M.W., K.D. and E.Ž.-Š. Provided feedback on the manuscript: L.A., A.C., P.G.-F., F.H., H.P., B.S., J.S., M.V. and V.V. All authors provided final approval of the version to be published; all authors agree to be personally accountable for the author's own contributions and for ensuring that questions related to the accuracy or integrity of any part of the work, even ones in which the author was not personally involved, are appropriately investigated, resolved and documented in the literature.

## Competing interests

The authors declare no competing interests.

## Additional information

**Correspondence and requests for materials** should be addressed to Kenny R. Coventry.

[1]School of Psychology, University of East Anglia, Norwich, UK. [2]Helmholtz Institute, Department of Experimental Psychology, Utrecht University, Utrecht, the Netherlands. [3]Hanse-Wissenschaftskolleg, Delmenhorst, Germany. [4]Department of English, Friedrich-Schiller-Universität Jena, Jena, Germany. [5]Department of Spanish, Modern and Classic Philology, University of the Balearic Islands, Palma, Spain. [6]Language Acquisition and Language Processing Lab, Department of Language and Literature, Norwegian University of Science and Technology, Trondheim, Norway. [7]Department of Spanish Language, Linguistics and Literature Theory, University of Seville, Seville, Spain. [8]Institute of Estonian and General Linguistics, University of Tartu, Tartu, Estonia. [9]Centre for the Arts in Society, Leiden University, Leiden, the Netherlands. [10]Võro Institute, Võru, Estonia. [11]Department of Linguistics, University at Buffalo, Buffalo, NY, USA. [12]Danieli Telerobot Srl, Genoa, Italy. [13]Istituto Italiano di Tecnologia (IIT), Genoa, Italy. [14]Paisii Hilendarski University of Plovdiv, Plovdiv, Bulgaria. [15]Department of English, University of Nevada, Reno, NV, USA. [16]Department of Slavonic Languages and Caucasus Studies, Friedrich-Schiller-Universität Jena, Jena, Germany. [17]Cognitive Science, Department of Humanities, Social and Political Sciences, ETH Zürich, Zürich, Switzerland. [18]Cognitive Science and Psycholinguistics Lab, Central Department of Linguistics, Tribhuvan University, Kathmandu, Nepal. [19]Centre for Neural and Cognitive Sciences, School of Medical Sciences, University of Hyderabad, Hyderabad, India. [20]Department of Communication and Cognition, TiCC, Tilburg University, Tilburg, the Netherlands. [21]Max Planck Institute for Psycholinguistics, Nijmegen, the Netherlands. [22]Department of Chemistry, University of Liverpool, Liverpool, UK. [23]Department of Computer Science, University of Manchester, Manchester, UK. [24]Laboratory for Perceptual and Cognitive Systems, Faculty of Computing, University of Latvia, Riga, Latvia. [25]Department of German, Friedrich-Schiller-Universität Jena, Jena, Germany. [26]School of Communication and Culture, Aarhus University, Aarhus, Denmark. [27]School of Languages and Translation Studies, University of Turku, Turku, Finland. [28]Institute for the Languages and Cultures of the Baltic, Vilnius University, Vilnius, Lithuania. [29]Centro de Investigación Nebrija en Cognición, Universidad Antonio de Nebrija, Madrid, Spain. [30]Department of Languages and Culture, Arctic University of Norway, Tromsø, Norway. [31]Interacting Minds Centre, Aarhus University, Aarhus, Denmark. [32]Centre for Humanities Computing, Department of Culture, Cognition and Computation, Aarhus University, Aarhus, Denmark. [33]Centre of Functionally Integrative Neuroscience, Aarhus University Hospital, Aarhus, Denmark. [34]Department of Computer Engineering, Bogazici University, Istanbul, Turkey. ✉e-mail: k.coventry@uea.ac.uk

# Reporting Summary

## Statistics

For all statistical analyses, confirm that the following items are present in the figure legend, table legend, main text, or Methods section.

| n/a | Confirmed | |
|---|---|---|
| ☐ | ☒ | The exact sample size ($n$) for each experimental group/condition, given as a discrete number and unit of measurement |
| ☐ | ☒ | A statement on whether measurements were taken from distinct samples or whether the same sample was measured repeatedly |
| ☐ | ☒ | The statistical test(s) used AND whether they are one- or two-sided<br>*Only common tests should be described solely by name; describe more complex techniques in the Methods section.* |
| ☒ | ☐ | A description of all covariates tested |
| ☐ | ☒ | A description of any assumptions or corrections, such as tests of normality and adjustment for multiple comparisons |
| ☐ | ☒ | A full description of the statistical parameters including central tendency (e.g. means) or other basic estimates (e.g. regression coefficient) AND variation (e.g. standard deviation) or associated estimates of uncertainty (e.g. confidence intervals) |
| ☐ | ☒ | For null hypothesis testing, the test statistic (e.g. $F$, $t$, $r$) with confidence intervals, effect sizes, degrees of freedom and $P$ value noted<br>*Give P values as exact values whenever suitable.* |
| ☒ | ☐ | For Bayesian analysis, information on the choice of priors and Markov chain Monte Carlo settings |
| ☐ | ☒ | For hierarchical and complex designs, identification of the appropriate level for tests and full reporting of outcomes |
| ☐ | ☒ | Estimates of effect sizes (e.g. Cohen's $d$, Pearson's $r$), indicating how they were calculated |

*Our web collection on statistics for biologists contains articles on many of the points above.*

## Software and code

Policy information about availability of computer code

| Data collection | Data collection was by hand - no software was used. |
|---|---|
| Data analysis | Data were analysed using bi- and multinomial multilevel modelling in SPSS version 27. |

For manuscripts utilizing custom algorithms or software that are central to the research but not yet described in published literature, software must be made available to editors and reviewers. We strongly encourage code deposition in a community repository (e.g. GitHub). See the Nature Portfolio guidelines for submitting code & software for further information.

## Data

Policy information about availability of data

All manuscripts must include a data availability statement. This statement should provide the following information, where applicable:
- Accession codes, unique identifiers, or web links for publicly available datasets
- A description of any restrictions on data availability
- For clinical datasets or third party data, please ensure that the statement adheres to our policy

All data and analysis scripts are available online with a url provided (also stated in the main manuscript): https://osf.io/ush2w/?view_only=1f38fa7ae6ce4bbab456eee80615ebe4

# Human research participants

Policy information about <u>studies involving human research participants and Sex and Gender in Research.</u>

| | |
|---|---|
| Reporting on sex and gender | The number of men and women are reported in the main manuscript (self-reported by participants); Supplementary Information Table S1 shows the number of males and females tested for each language. A priori, as stated in the main manuscript, researchers at all sites set out to test an equal number of men and women (self-reported). For the analyses we did not include sex as a variable given the goal of study was to consider general patterns across languages and the extent of variation across all participants within languages. The data are available (open source) for future analyses of data by sex. |
| Population characteristics | Demographic information is provided in Supplementary Information Table S1 (the mean age of the sample was 26 years, SD = 7.64). As the goal of study was to consider general patterns across languages and the extent of variation across all participants within languages, we did not examine age as a potential predictor of differences in demonstrative use. The data are available (open source) for future analyses of data by age |
| Recruitment | Participants took part either for nominal payment, course credit, or on a voluntary basis (commensurate with cultural norms of participation for each language). The lead researcher at each language site was responsible for recruiting participants in a culturally appropriate manner. Participants were all volunteers recruited through local advertising, word of mouth, etc. commensurate with norms for recruitment at each site. Given that participants were blind to the purpose of the study and were all native speakers (L1 speakers from birth) of the languages tested, self-selection bias is unlikely to have affected the integrity and representativeness of the data. |
| Ethics oversight | Prior to data collection, the study received full ethical clearance from the University of East Anglia's School of Psychology Ethics Committee (approval numbers 13-14-5 and 2017-0034-000748 granted respectively on 9/3/2015 and 8/9/2017) covering data collection across languages. Local clearance was also required for Finnish data collection (from Tartu University, approval number 293/T-21, granted on 20/5/2019). All procedures were carried out in accordance with the guidelines of the BPS, APA, APS and the Declaration of Helsinki. |

Note that full information on the approval of the study protocol must also be provided in the manuscript.

# Field-specific reporting

Please select the one below that is the best fit for your research. If you are not sure, read the appropriate sections before making your selection.

☐ Life sciences    ☒ Behavioural & social sciences    ☐ Ecological, evolutionary & environmental sciences

For a reference copy of the document with all sections, see <u>nature.com/documents/nr-reporting-summary-flat.pdf</u>

# Behavioural & social sciences study design

All studies must disclose on these points even when the disclosure is negative.

| | |
|---|---|
| Study description | The study is a cross-linguistic experimental quantitative study using the 'memory game method' pioneered by the lead author. The experiment measures choice of demonstrative in each of 29 languages with varied demonstrative systems manipulating the distance the reference object is from the speaker (participant) and the position of the addressee. |
| Research sample | Languages were selected in accordance with four working criteria: i) sampling across languages with demonstrative systems varying in number of demonstrative terms, ii) sampling between and within language families, iii) sampling across geographical areas, and iv) the availability of researchers to collect data in targeted languages. The sample of 29 languages spans geographical areas, genetic origins, and differences in spatial communication systems.<br><br>Participants for each language were all L1 speakers of the language tested. |
| Sampling strategy | A statistical power analysis (a priori) was performed for sample size estimation using G*Power. With power = 0.9 and an alpha = .05, and the effect sizes reported in Coventry et al.32, the projected sample size is approximately N = 17 for each language. Given the effect size was based on English, and that many languages tested have no empirical data on demonstrative production, 17 participants was set as a minimum sample size, while aiming for 30+ per language (N = 914, M = 32 participants per language).<br><br>Convenience sampling was used, while ensuring an equal balance (where possible) of male and female participants for each language (self reported). The age range of participants was broadly equivalent across languages (see Supplementary Information Table S1). Given large cultural differences across language samples, researchers were sensitive to local norms regarding the conduct of experimental work. |

| Data collection | The experiment employed the 'memory game' method (see references 34-36, 52, main manuscript). This established method was designed to elicit language under strictly controlled experimental conditions, but without participants being aware that language data were being collected. To do so, participants were instructed that they were taking part in a study on the effects of language on object-location memory, with memory probe trials maintaining the cover of the memory experiment throughout (see 52 for more detail).

Participants were seated at a large table (325cm long), with 12 marked locations (colored dots), spaced equidistantly 25cm apart down the midline of the table directly in front of participants, starting at 25cm from the participants' edge of the table (Supplementary Information Figure S1). On each trial, an object was placed by the experimenter (the 'addressee') on one of the colored dots. The objects placed were colored shapes on disks. All shapes were basic geometric shapes: a black cross/green star/ yellow triangle/red circle/blue heart/orange square/red sun/white moon/red moon/black bar. In each language 6 objects from the set were selected (4 for Tseltal, given available colour terms), ensuring the language had a color lexicon able to differentiate the colors, and ensuring all objects were matched for gender (in gendered languages).

The experiment manipulated the distance of the object from the participant (the 'speaker' in the experiment), and the position of the addressee. The addressee was seated either next to or opposite the speaker. This addressee position manipulation was blocked and counterbalanced: the addressee switched their position once, halfway through the experiment. The distance condition was pseudo-randomized, to ensure no object or distance was used in two successive trials, preventing carry-over effects. 6 of the marked locations on the table were used, creating 3 conceptual regions: Region 1 - within the speaker's reach/peripersonal space (PPS), at 25cm and 50cm; Region 2 - out of reach for both speaker and addressee (regardless of addressee position) and at medium distance from the speaker, at 150cm and 175cm; Region 3 - at 275cm and 300cm furthest from the speaker, but in the PPS of the addressee when the addressee was seated opposite the participant.

(This information is in the main manuscript.)

Demonstrative choices were recorded using pencil and paper (ticking the demonstrative used on each given trial).

Researchers collecting the data were aware of the main manipulations in the study, but the method eliminated any potential for researchers to bias the outcome of the study. |
|---|---|
| Timing | As stated in the main manuscript. data were collected between January 2016 and December 2019, with staggered tested of languages during that period (so all testing sites used the same apparatus). |
| Data exclusions | Of the 914 tested, data from 40 participants were excluded based on the following a priori criteria: a) participants did not have normal or corrected to normal vision, b) participants guessed that the study was about demonstrative use, c) participants reported deliberately using demonstratives in a way they wouldn't normally use them in the study. All data exclusions took place prior to the data being submitted to independent statistical analyses. |
| Non-participation | No participants declined - participation was on a voluntary basis. |
| Randomization | The experiment was repeated measures so randomization of participants to conditions does not apply. However, trial order was psuedo-randomized (see Data Collection section above) |

# Reporting for specific materials, systems and methods

We require information from authors about some types of materials, experimental systems and methods used in many studies. Here, indicate whether each material, system or method listed is relevant to your study. If you are not sure if a list item applies to your research, read the appropriate section before selecting a response.

## Materials & experimental systems

| n/a | Involved in the study |
|---|---|
| ☒ ☐ | Antibodies |
| ☒ ☐ | Eukaryotic cell lines |
| ☒ ☐ | Palaeontology and archaeology |
| ☒ ☐ | Animals and other organisms |
| ☒ ☐ | Clinical data |
| ☒ ☐ | Dual use research of concern |

## Methods

| n/a | Involved in the study |
|---|---|
| ☒ ☐ | ChIP-seq |
| ☒ ☐ | Flow cytometry |
| ☒ ☐ | MRI-based neuroimaging |

