## [Peer Review File · Nature Human Behaviour]

Peer Review Information

Journal: Nature Human Behaviour

Manuscript Title: Spatial Communication Systems Across Languages Reflect Universal Action Constraints

Corresponding author name(s): Kenny R. Coventry

Reviewer Comments & Decisions:

Decision Letter, initial version:
--

21st November 2022

Dear Professor Coventry,

Thank you once again for your manuscript, entitled "Spatial Communication Systems Across Languages Reflect Universal Action Constraints", and for your patience during the peer review process.

Your Article has now been evaluated by 3 referees. You will see from their comments copied below that, although they find your work of considerable potential interest, they have raised quite substantial concerns. In light of these comments, we cannot accept the manuscript for publication in the current form, but we would be interested in considering a revised version if you are willing and able to fully address reviewer and editorial concerns.

We hope you will find the referees' comments useful as you decide how to proceed. If you wish to submit a substantially revised manuscript, please bear in mind that we will be reluctant to approach the referees again in the absence of major revisions to address all of the points raised by the reviewers.

In particular, you will see that Referee #2 has significant concerns over whether the data presented here are able to answer the important questions around spatial demonstratives, because the current experiment only manipulated spatial distance which is only one of the factors of potential importance. Reviewer #1 also raises similar concerns. We believe that a compelling response to these concerns is required, if needed with additional data and analysis (either analysis of new or existing data). Please contact me if you wish to discuss these revisions.

If you wish to submit a suitably revised manuscript we would hope to receive it within 4 months. I would be grateful if you could contact us as soon as possible if you foresee difficulties with meeting this target resubmission date.

- Include a "Response to the editors and reviewers" document detailing, point-by-point, how you addressed each editor and referee comment. If no action was taken to address a point, you must provide a compelling argument. When formatting this document, please respond to each reviewer comment individually, including the full text of the reviewer comment verbatim followed by your response to the individual point. This response will be used by the editors to evaluate your revision and sent back to the reviewers along with the revised manuscript.
- Highlight all changes made to your manuscript or provide us with a version that tracks changes.

[REDACTED]

Thank you for the opportunity to review your work. Please do not hesitate to contact me if you have any questions or would like to discuss the required revisions further.

Sincerely,
Jamie

Dr Jamie Horder
Senior Editor
Nature Human Behaviour

REVIEWER COMMENTS:

Reviewer #1:
Remarks to the Author:

This paper undertakes an experiment, across 29 diverse languages, of demonstrative words like this/that. The setup is there is a speaker sitting at a table (with a listener either next to them or on

the other side of the table). They are prompted to refer to objects either near, in the middle ground, or far. They find that the largest source of distinction is whether the object can be touched or not. Some languages show listener-specific effects, such that the position of the listener matters.

Undertaking a controlled study across 29 genuinely diverse languages is an impressive feat. Often typological studies rely on data in grammars. And when there are experiments run, they are often simple surveys (as in the World Color Survey). This is a full experiment run in a very wide range of languages. The experiment is sound, and it addresses a major question: the relationship between cognitive/spatial universals and linguistic structure across languages. As such, it is likely to be impactful and of wide interest in the language and cognition communities. Thus, I believe it is appropriate for a high-profile, general journal Nature Human Behavior and I hope to see it published.

I do, however, have several questions, as well as suggestions for how the paper could be better, which I believe should be addressed before publication. Because these are substantive, I would recommend revision before it should be accepted.

First: There is a strong claim made that the experiment provides evidence that being within reach is the defining criterion for the proximal category. But it also seems like in this experiment the readiness to hand is confounded with distance. If there was some non-linear distance decay function that determined distance, would that generate the same result? That is, how do we know it's just about readiness to hand and not distance more generally?

Second: as stated, I think the data and experiment are important and illuminating, but the graphs and analyses underdeliver. In particular, there is clearly a lot to learn from this data about between-language differences, but the paper pays surprisingly little attention to these. The introduction sets up big questions about listener-oriented vs. speaker-oriented languages, but the graphs in Figs 3 and 4 are not broken down by those claims. And they both make it very hard to see what patterns emerge across languages. It seems particularly a shame that Figure 4 is organized alphabetically instead of by deictic system or by region or language family or some other meaningful category. I would much rather see the data (not the model output, but the raw data) plotted in meaningful categories to see whether, for instance, languages claimed to be listener-oriented all are different from so-called deictic-oriented languages, whether they all look alike etc. But these plots don't make that easy to see.

Moreover, Figure 3 makes it almost impossible to see the individual languages and so is disappointing for the same reason. The bottom plot has a couple outliers in the 1 column, but what languages are those and why?

Third: The experimental setup is clever, and it is nice that pains are taken to make sure the experimenters are unaware of the goal of the experiment. (There is a memory game tacked on top.) But I would have appreciated more detail what is meant by "they were asked to point at and name the object placed on the table, using three words." How many examples were given? There is an obvious concern that participants could be biased towards referring to some objects in a particular way (e.g., if the demo involves "this" and a proximal object, that could be stacking the deck in a particular way that might be different if the demo involved "that" and a distal one). How was this part of the procedure done?

These are all issues that I think are important. But I do think they can be addressed. And, as stated at the beginning, I believe this will be important and impactful work.

Reviewer #2:

Remarks to the Author:

Using an experimental paradigm, this study explores the spatial usage of proximal/distal demonstrative distinctions across 29 languages. It identifies a cross-linguistically stable mapping of such distinctions, where proximal forms associate with referents that are reachable by the speaker. It concludes that this association is universal and driven by fundamentally spatial constraints on speaker action in relation to object referents. It further argues, contrary to some current theories of spatial language and cognition, that there are universal features of spatial categorization and that the class of demonstratives shows that a body-based, egocentric perspective is central to representations of space. Extending experimental 'memory game' methodology previously applied to English demonstratives, it is the first major experimental study to explore demonstrative distinctions quantitatively across a large number of languages. This makes it rather impressive in scope. However, I have a few concerns regarding the conceptual and methodological basis of the study, outlined below:

First, the paper does not quite make it clear what it sets out to test, or what its results entail. The abstract, significance statement and discussion section make brief reference to "semantic universals", but the paper never explicitly states that this is what it investigates. Nor does it explicitly claim that this is what it has found. It is clearer about having tested and identified "universal constraints" (on action) but how is this to be understood in linguistic terms? That a speaker's ability to reach/act on an object is encoded semantically in the demonstrative system of every language on the planet? Or do the constraints apply at some other level, leading to cross-linguistically stable but epiphenomenal (perhaps even experiment specific) patterns in speakers' deployment of demonstrative contrasts? I'm assuming the authors claim the former, but this should then be properly clarified and justified.

My second and more serious concern is the relationship between the general debate on demonstrative function, the goals of the study, and its experimental design. As the authors point out, the main theoretical point of contention in demonstrative research is whether demonstratives are primarily a spatial, egocentric and body-oriented strategy of reference, or an interactionally driven strategy dedicated to manipulating the relationship between speaker, addressee, and referent (lines 138-143). While recognizing that the two approaches are not mutually exclusive, the paper squarely sides with the former in terms of what is considered the most fundamental dimension in demonstrative function. Curiously, though, the experiment does not address this tension between spatial and interactional explanations. It is purely designed to probe select spatial parameters, primarily the referent's distance from the speaker on the sagittal axis, and secondarily the addressee's location (face-to-face vs side-by-side with speaker). The main parameters of the interactional dimension – those pertaining to the addressee's epistemic and attentional footing in relation to the referent (not to be confused with the relative positions of speaker and addressee) – are not at all manipulated in the experimental design. Thus, we do not get to know if the observed spatial patterns would stand their ground in the face of such competing parameters, and the big question remains unanswered. In fact, as far as I can understand, the addressee's attentional/epistemic relationship to the referent remains the same in every trial, and I'm wondering if this constant might somehow contribute to the observed spatial

patterns. It is particularly noteworthy that the addressee was consistently represented by the experimenter, who one would assume was instructed to behave identically in every trial and who from the participant speaker's perspective would have been considered all-knowing about the referent. Again, I'm concerned about epiphenomena that are at odds with the paper's main claims. At the very least, the authors should explain and justify their reasons for not considering such significant competing parameters, and how that might influence the results.

A third point concerns participant instructions. The paper states that its method "avoids participants consciously choosing to use demonstratives during the task" (lines 211-212). At the same time, participant instructions seem to have been very explicit about them. The procedure description states that participants were instructed to use three words ("[demonstrative] [object color] [object shape name]", lines 498-500). Given that these instructions must have involved an example of a demonstrative form, how did the researchers ensure that the instructions did not influence demonstrative choice? Further on, the procedure description says "Another key check was that participants across sites were instructed to use all demonstrative forms available to them in their language" (lines 528-529). To me this all suggests that participants went through some fairly significant metalinguistic priming about demonstratives pre-trial. The memory game format may well have helped towards neutralizing this, and the debriefing procedure is somewhat reassuring, but I find it very hard to believe that participants were not consciously choosing to use demonstratives during the task given that they were explicitly asked to do so. I think this aspect of the procedure needs to be further explained and problematized.

Finally a comment about the language sample: The wording suggests the study aimed for a diverse sample of languages (lines 222-230); however, there is still a considerable genealogical and geographical bias in the sample: 13 of the 29 languages belong to the Indo-European family, 18 are spoken in western Eurasia. This may have been difficult to avoid and is not necessarily a problem, but I think it should be acknowledged that there are limits to the diversity that this sample captures.

Reviewer #3:

(Please see attached PDF)

Nature Human Behaviour — 25 October 2022
Spatial communication systems across languages reflect universal action constraints
 K.R. Coventry *et al.*

The submission makes a robust and welcome contribution to work on demonstrative systems and their place in linguistic universals, presenting a readily replicable procedure for determining how the position of speaker, hearer, and object determine the use of demonstratives in a given language (for a given speaker). It could be published as is. However, as the paper stands, it misses the opportunity to engage with some sympathetic research agendas and hence narrows its readership. I suggest broadening its appeal to maximize the paper's impact. A selection of points follows, most of them on this theme but some of them raising empirical questions.

The opening sentence of the Significance Statement is (lines 92–93): “Key to the debate regarding the very nature of language is the existence or absence of language universals.” To my mind, this sets off on the wrong foot. Putting my cards on the table, I am a fieldworker, a theoretical linguist, and a typologist. In none of those guises do I think there is much life in the hunt for surface-true universals. As a typologist, I look for confluences of factors leading to crosslinguistic tendencies. As a theoretician, the universals that interest me are the building blocks and building processes that create linguistic constructions. And as a fieldworker, I’m interested in problem cases for both. The issue with the opening sentence is not whether the paper finds a universal but whether the search for universals is key to understanding language. I suggest the authors set the discussion up differently. (E.g.: “Spatial communication has been argued to be a domain in which semantic universals, if they exist, should be apparent. The present paper”)

I further question whether 29 languages is enough to serve as the basis for any claimed universal. For instance, Bhat 2004 uses well over 200 (my books are in movers’ boxes so I can’t check the exact figure). The issue here is not with the content of the paper but with how the significance of its finding is contextualized. The significance, to my mind, is that the paper finds a potential universal building block amongst geographically and typologically diverse demonstrative systems. Responses to Evans and Levinson 2009 homed in on that work’s preoccupation with the straw man of surface-true universals instead of what we might call ‘building block universals’. The paper fits naturally in with the more abstract view of universals, particularly in light of how it ties its findings to potential action by the speaker (and hearer).

In several places (e.g., lines 180–190, 319), the paper refers to the small number of participants in field-based studies. Line 319 points the finger at anthropologists. Where does this leave descriptive linguists, who might either in communities where the language is ubiquitous or in ones where the language is moribund and rarely used even for basic conversations? Doing fieldwork in the latter circumstance, one does indeed suffer from small sample sizes. In the former circumstance, however, the language is all around and you constantly readjust your understanding based on everything from intensive elicitation sessions to eavesdropping at bus stops. This ubiquitous scenario does not appear to be acknowledged in the paper but it should be. For linguists in the sporadic-use scenario, the paper could usefully suggest how many judgments from how many consultants fieldworkers require to get to grips with

a demonstrative system. This, again, strikes me as a point where the paper misses out on engaging (in a very practical way) with a community of researchers.

Recent theoretical linguistic work seems to me to be sympathetic to the paper’s point of view. Harbour 2016 and Ackema and Neeleman 2018 are analyses of person systems which take demonstrative systems to be, in essence, at root person systems. In three*-term demonstrative systems, Harbour draws a distinction between those in which the medial term is addressee based and those in which it is distance based. This seems to map onto distinction discussed in lines 302–314.

The discussion of two-term systems in the same paragraph is both revealing and, again, a possible point at which to engage with descriptive linguists. Bulgarian is, along with Catalan, a well cited case in which the ‘speaker-centred’ demonstrative extends into the hearer’s space (see Imai 2003 for references; they are central to the works in the last paragraph). I was surprised to see Italian and Mandarin in there too. Maybe I just don’t know the descriptive literature well enough but it is possible that the authors have made a discovery here that traditional descriptive grammars have ignored. If so, the discussion should again engage with other researchers, implicitly inviting them to revisit and revise traditional descriptions.

Satawal has been described (Yoshida, 1981) as distinguishing in-hand/reachable from near to speaker. The paper implies a reduction of 'near to speaker' to reachable. Is Satawal therefore a potential problem?

The notion, expressed in the paper's title and also in line 408, that we conceive of the world in terms of how we can act on it reminded me of Dixon's 1994 study of ergativity. (Again, sorry that I can't provide a page reference as the book is boxed up.) The authors may wish to cite this.

References

- Ackema, Peter and Neeleman, Ad. 2018. *Features of Person: From the Inventory of Persons to Their Morphological Realization*. Cambridge MA: MIT Press.
- Bhat, D. N. S. 2004. *Pronouns: A Cross-Linguistic Study*. Oxford: Oxford University Press.
- Dixon, R.M.W. 1994. *Ergativity*. Cambridge: Cambridge University Press.
- Evans, Nicholas and Levinson, Stephen. 2009. The myth of language universals: Language diversity and its importance for cognitive science. *Behavioral and Brain Sciences* 32:499–499.
- Harbour, Daniel. 2016. *Impossible Persons*. Cambridge, MA: MIT Press.
- Imai, Shingo. 2008. *Spatial deixis*. Ph.D. thesis, State University of New York, Buffalo.
- Yoshida, Shuji. 1981. Kūkan ninshiki no ruikeika ni tsuite [on typology of spatial recognition]. *Kikan Jinruigaku* 12–13:80–129.

Author Rebuttal to Initial comments

Spatial Communication Systems Across Languages Reflect Universal Action Constraints: Response to Reviewers' Comments

We would like to thank the three reviewers for careful consideration of our manuscript. Attached is a revised version of the main manuscript and supplementary information that addresses the points raised. Below we address each point one-by-one (points raised by reviewers in bold italic, responses in plain text). Please note that the changes are marked in the text in yellow for ease of location in the revised manuscript and SI.

Reviewer 1

R1.1 *There is a strong claim made that the experiment provides evidence that being within reach is the defining criterion for the proximal category. But it also seems like in this experiment the readiness to hand is confounded with distance. If there was some non-linear distance decay function that*

determined distance, would that generate the same result? That is, how do we know it's just about readiness to hand and not distance more generally?

We thank the reviewer for raising this issue. In the previous version of the manuscript we acknowledged that, while the data are consistent with a distance effect, the pattern of data mitigates in favour of the reachable/nonreachable distinction based on a) the rapid fall off in proximal use beyond reachable distance, and b) consistency with results from previous studies. We have extended discussion at the same point in the paper to make it clearer why we argue that this is the case (please see first highlighted section beginning at the foot of page 5), while also referring readers to a new section in the SI – “Interrogating the distance effect” – that fully considers the argument that the data are more consistent with a reachable/nonreachable distinction than with a mere distance effect (please see new section in SI on pages 13 and 14). This includes short discussion of how space is encoded and an overview of studies that have tested the effects of reachability directly on demonstrative choice through tool use and varying pointing hand, allowing reachability to be disentangled from distance (mapping onto evidence for distinct brain systems processing peripersonal and extrapersonal space). Moreover, data from non-linguistic ‘memory game’ studies are considered alongside the present data, showing a similar drop off between proximal term use and distance memory error at the same distances. In sum, there are compelling reasons for arguing that the data support a reachable/nonreachable distinction, now clearly articulated in the text.

R1.2 Second: as stated, I think the data and experiment are important and illuminating, but the graphs and analyses underdeliver. In particular, there is clearly a lot to learn from this data about between-language differences, but the paper pays surprisingly little attention to these. The introduction sets up big questions about listener-oriented vs. speaker-oriented languages, but the graphs in Figs 3 and 4 are not broken down by those claims. And they both make it very hard to see what patterns emerge across languages. It seems particularly a shame that Figure 4 is organized alphabetically instead of by deictic system or by region or language family or some other meaningful category. I would much rather see the data (not the model output, but the raw data) plotted in meaningful categories to see whether, for instance, languages claimed to be listener-oriented all are different from so-called deictic-oriented languages, whether they all look alike etc. But these plots don't make that easy to see. Moreover, Figure 3 makes it almost impossible to see the individual languages and so is disappointing for the same reason. The bottom plot has a couple outliers in the 1 column, but what languages are those and why?

We thank the reviewer for constructive and helpful suggestions regarding presentation of figures. We have completely revised Figure 3 to show data for individual languages (bottom panel) whilst retaining a visual for the overall effect of distance across languages (top panel). We have also included a new Figure,

Figure 5, that displays plots for the eight languages with main effects of addressee position and/or interactions between addressee position and distance. The addition of Figure 5 together with the improved version of Figure 3 will allow readers interested in specific languages to pick out the data more clearly.

In theory we liked the suggestion from the reviewer that languages could be classified based on whether they are person-centred or not and mapped onto our data accordingly. However, in practice it is not the case that linguists agree for every language whether an individual language is person-centred or not, hence this was not feasible as a display option. So we have opted for individual language plots, clearly showing the languages that are two-term or three-term based on the number of demonstratives in each plot.

Regarding potential outliers in the data, one can see from Figure 3 that proximal terms in languages confirm to the general pattern, evidenced by the contrasts in the analyses for individual languages presented in the SI. However, we now note in the manuscript that languages often have neutral (Levinson, 2018) or default (Diessel & Monakhov, 2022) demonstratives that can be used for any location – for example, in English, *that* can be used for any location - so the proximal term overall is not used all the time within peripersonal space. Different languages have different default demonstratives, and some languages have stronger defaults than others. Hence there is some inter-individual variability in the data (as one would expect). We have added this point to the discussion (please see the first highlighted section on page 8).

R1.3 Third: *The experimental setup is clever, and it is nice that pains are taken to make sure the experimenters are unaware of the goal of the experiment. (There is a memory game tacked on top.) But I would have appreciated more detail what is meant by “they were asked to point at and name the object placed on the table, using three words.” How many examples were given? There is an obvious concern that participants could be biased towards referring to some objects in a particular way (e.g., if the demo involves “this” and a proximal object, that could be stacking the deck in a particular way that might be different if the demo involved “that” and a distal one). How was this part of the procedure done?*

As the reviewer recognises, the ‘memory game’ method was designed to avoid many of the pitfalls of asking participants to use language as a primary goal of the task (see also response to Reviewer 2, below). The procedure (as published open access in JOVE), involves no examples of the use of specific

demonstratives by experimenters, and no examples of specific demonstratives in the instructions tied to specific locations. The procedure is carefully designed to avoid ‘stacking the deck’ or priming participants to use demonstratives in particular ways. This is now made clearer in the manuscript, with the addition of a paragraph explaining the protocol in more detail in the methods section of the paper (please see highlighted section on page 10).

Reviewer 2

R2.1 First, the paper does not quite make it clear what it sets out to test, or what its results entail. The abstract, significance statement and discussion section make brief reference to “semantic universals”, but the paper never explicitly states that this is what it investigates. Nor does it explicitly claim that this is what it has found. It is clearer about having tested and identified “universal constraints” (on action) but how is this to be understood in linguistic terms? That a speaker’s ability to reach/act on an object is encoded semantically in the demonstrative system of every language on the planet? Or do the constraints apply at some other level, leading to cross-linguistically stable but epiphenomenal (perhaps even experiment specific) patterns in speakers’ deployment of demonstrative contrasts? I’m assuming the authors claim the former, but this should then be properly clarified and justified.

We thank the reviewer for raising this point. We have revised the article making it clearer what is being tested and what claims we think are warranted from the data and analyses. In particular, at the point in the paper where we set out the goals of the study, we now explicitly state that we are testing for semantic universals (please see highlighted section on page 4). And in the discussion section of the paper we now explicitly state that the data are consistent with the presence of semantic universals, with specific terms in each language mapping onto a reachable/nonreachable distinction (please see highlighted section towards the bottom of page 6).

R2.2 My second and more serious concern is the relationship between the general debate on demonstrative function, the goals of the study, and its experimental design. As the authors point out, the main theoretical point of contention in demonstrative research is whether demonstratives are primarily a spatial, egocentric and body-oriented strategy of reference, or an interactionally driven strategy dedicated to manipulating the relationship between speaker, addressee, and referent (lines 138-143). While recognizing that the two approaches are not mutually exclusive, the paper squarely sides with the former in terms of what is considered the most fundamental dimension in demonstrative function. Curiously, though, the experiment does not address this tension between spatial and interactional explanations. It is purely designed to probe select spatial parameters,

primarily the referent's distance from the speaker on the sagittal axis, and secondarily the addressee's location (face-to-face vs side-by-side with speaker). The main parameters of the interactional dimension – those pertaining to the addressee's epistemic and attentional footing in relation to the referent (not to be confused with the relative positions of speaker and addressee) – are not at all manipulated in the experimental design. Thus, we do not get to know if the observed spatial patterns would stand their ground in the face of such competing parameters, and the big question remains unanswered. In fact, as far as I can understand, the addressee's attentional/epistemic relationship to the referent remains the same in every trial, and I'm wondering if this constant might somehow contribute to the observed spatial patterns. It is particularly noteworthy that the addressee was consistently represented by the experimenter, who one would assume was instructed to behave identically in every trial and who from the participant speaker's perspective would have been considered all-knowing about the referent. Again, I'm concerned about epiphenomena that are at odds with the paper's main claims. At the very least, the authors should explain and justify their reasons for not considering such significant competing parameters, and how that might influence the results.

The manipulations in the experiment were chosen precisely because they have been identified in the literature on demonstratives as the most key distinctions that demonstrative systems are hypothesised to make, while getting to the heart of the debate regarding egocentric versus interactionally-driven use. We do not agree with the reviewer that the manipulation of addressee position represents a purely spatial manipulation. Rather, the manipulation of the position of the addressee was motivated as the key interactional factor that has been posited in the literature (a point we make in the manuscript).

The idea is that face-to-face interaction entails taking the attention of the addressee into account, conflicting with the view that demonstrative use is egocentric. Rather than being purely spatial, when the addressee is seated opposite the speaker the joint attentional episode is qualitatively different to the case where speaker and addressee are side-by-side. The speaker must take into account where objects are with respects to both themselves *and* the addressee. As such, theory of mind is implicated in this setting (see for example Rubio-Fernandez, 2020), in that the speaker has to be aware of the attentional orientation of the addressee with respect to both the referent and herself/himself. This leads to a range of possible predictions from the literature regarding how addressee/speaker misalignment of perspective impacts demonstrative choice. And it is these predictions, as set out in the paper (e.g. Figure 1), that are at stake here. One prediction is that some languages take the perspective of the addressee into account directly when using demonstratives (so called person-centred languages), and others do not. Another view is that demonstratives in the face-to-face setting introduces the idea of a shared (social) space between the speaker and hearer, licensing the use of the proximal term at any distance

between the speaker and hearer (please see the cited referenced within the body of the manuscript). So the addressee manipulation should not be regarded as merely spatial.

A rather different question, and one we think the reviewer might be alluding to, is the case where joint attention is *not* apparent prior to demonstrative use, as in the case where the addressee is looking away, for example. Such a case has been examined in Turkish (as we note in the manuscript) where there is evidence that the medial term (*şu*) has a (part) function to be used when the addressee is not paying attention to the space between speaker and hearer. Indeed, Küntay and Özyürek (2006) and Rubio-Fernandez (2022) have shown that when an addressee is attentionally disengaged (looking away) from an interaction, the term serves as a directive for attention correction. However, we note that such a finding was not found for Japanese (Rubio-Fernandez, 2022), suggesting that specific terms with an attention reorienting function may only be present in some three-term languages. Moreover, while demonstrative use during disengaged attentional episodes is important, it has to be considered in the broader context of deictic communication. In most accounts of deictic communication joint attention is regarded as a precondition for demonstrative use (Diessel, 2006; Kita, 2003). The evidence to date, mainly from developmental studies, indicates that joint attention is usually in place prior to demonstrative use. For example, Todisco and colleagues (2021) analysed attention immediately prior to 518 deictic episodes involving Italian caregivers and infants (aged 1;08 – 2;07) in story book interactions. Just prior to the production of a demonstrative by either a caregiver or infant, both caregiver and infant were engaged in joint attention in the vast majority of deictic episodes (c. 85%). So while demonstratives are sometimes used in cases where an addressee is not attentionally engaged, such instances are arguably less central to deictic communication than the case where joint attention is in place prior to the speaker drawing the addressee's attention towards a specific object.

Given the use of an experimental paradigm with sufficient numbers of trials and participants to be powerful enough to generalise to the population for each language, we had to restrict the number of variables tested. At the inception of the project, we discussed the range of factors in the literature that have been posited to affect demonstrative choice across languages, and chose the variables that are most prominent in the literature, and at the same time would help to disentangle the extent to which demonstrative use across languages is based on egocentric spatial use versus a more 'sociocentric' approach taking into consideration the perspective of the addressee. Attention disengagement and elevation were two variables that were next on the list, and indeed future work would do well to examine these.

In the revision we have been careful to acknowledge that there are other manipulations that will be exciting to test in future studies, and we now specifically identify the issue of disengaged attention as an example of such a manipulation (please see the middle highlighted section on page 8). We have also made it clear earlier in the manuscript that joint attention usually precedes spatial demonstrative use (please see the highlighted section at the bottom of page 2), which will make readers aware that the focus of our attention is on such normative episodes.

R2.3 A third point concerns participant instructions. The paper states that its method “avoids participants consciously choosing to use demonstratives during the task” (lines 211-212). At the same time, participant instructions seem to have been very explicit about them. The procedure description states that participants were instructed to use three words (“[demonstrative] [object color] [object shape name]”, lines 498-500). Given that these instructions must have involved an example of a demonstrative form, how did the researchers ensure that the instructions did not influence demonstrative choice? Further on, the procedure description says “Another key check was that participants across sites were instructed to use all demonstrative forms available to them in their language” (lines 528-529). To me this all suggests that participants went through some fairly significant metalinguistic priming about demonstratives pre-trial. The memory game format may well have helped towards neutralizing this, and the debriefing procedure is somewhat reassuring, but I find it very hard to believe that participants were not consciously choosing to use demonstratives during the task given that they were explicitly asked to do so. I think this aspect of the procedure needs to be further explained and problematized.

In the revision we have more carefully set out the memory game procedure to make it clear that priming did not occur (please see highlighted section on page 10; see also response to Reviewer 1 above). In the instructions participants were always given all the forms in the language from which to freely choose, and examples were never given involving a specific demonstrative at a specific location. The point about “consciously choosing” contrasts the memory game method - where demonstrative use is incidental to the task - with methods involving linguistic informants who are often asked explicitly to think about how they use specific terms and are therefore thinking about how they use them as an explicit part of the elicitation process. In the present method the secondary nature of language changes the task from one where participants know the experimenter/linguist wants to know how they use specific words in their language to one where the focus is on memory. It is certainly the case that different methods have different strengths and weaknesses – a point we now discuss (please see the bottom highlighted section on page 8; see also the response to Reviewer 3) – but the method we employed does afford experimental control over what is tested thereby providing reassurance that effects are not a consequence of demand characteristics.

R2.4 Finally a comment about the language sample: The wording suggests the study aimed for a diverse sample of languages (lines 222-230); however, there is still a considerable genealogical and geographical bias in the sample: 13 of the 29 languages belong to the Indo-European family, 18 are spoken in western Eurasia. This may have been difficult to avoid and is not necessarily a problem, but I think it should be acknowledged that there are limits to the diversity that this sample captures.

This point is well taken. Limits to the diversity of the sample are now acknowledged in the revision (please see the bottom highlighted section on page 8). Moreover, as the method is open source, we hope that the method can be used by others in the field to test speakers of other diverse languages.

Reviewer 3

R3.1 The opening sentence of the Significance Statement is (lines 92–93): “Key to the debate regarding the very nature of language is the existence or absence of language universals.” To my mind, this sets off on the wrong foot. Putting my cards on the table, I am a fieldworker, a theoretical linguist, and a typologist. In none of those guises do I think there is much life in the hunt for surface-true universals. As a typologist, I look for confluences of factors leading to crosslinguistic tendencies. As a theoretician, the universals that interest me are the building blocks and building processes that create linguistic constructions. And as a fieldworker, I’m interested in problem cases for both. The issue with the opening sentence is not whether the paper finds a universal but whether the search for universals is key to understanding language. I suggest the authors set the discussion up differently. (E.g.: “Spatial communication has been argued to be a domain in which semantic universals, if they exist, should be apparent. The present paper”)

We thank the reviewer for this suggestion. Indeed, there was discussion among the authors regarding the thorny issue of universals identified by the reviewer. We have revised the opening sentence of the significance statement accordingly. We have also revised the Abstract accordingly.

R3.2 I further question whether 29 languages is enough to serve as the basis for any claimed universal. For instance, Bhat 2004 uses well over 200 (my books are in movers’ boxes so I can’t check the exact figure). The issue here is not with the content of the paper but with how the significance of its finding is contextualized. The significance, to my mind, is that the paper finds a potential universal building block amongst geographically and typologically diverse demonstrative systems. Responses to Evans

and Levinson 2009 homed in on that work's preoccupation with the straw man of surface-true universals instead of what we might call 'building block universals'. The paper fits naturally in with the more abstract view of universals, particularly in light of how it ties its findings to potential action by the speaker (and hearer).

We thank the reviewer for this point. We have added a sentence to the discussion recognising the limitation of the language sample (please see the bottom highlighted section on page 8; see also the response to Reviewer 2 above). We have also made some changes to how the finding is contextualised as suggested in the discussion (see highlighted sections).

R3.3 In several places (e.g., lines 180–190, 319), the paper refers to the small number of participants in field-based studies. Line 319 points the finger at anthropologists. Where does this leave descriptive linguists, who might either in communities where the language is ubiquitous or in ones where the language is moribund and rarely used even for basic conversations? Doing fieldwork in the latter circumstance, one does indeed suffer from small sample sizes. In the former circumstance, however, the language is all around and you constantly readjust your understanding based on everything from intensive elicitation sessions to eavesdropping at bus stops. This ubiquitous scenario does not appear to be acknowledged in the paper but it should be. For linguists in the sporadic-use scenario, the paper could usefully suggest how many judgments from how many consultants fieldworkers require to get to grips with a demonstrative system. This, again, strikes me as a point where the paper misses out on engaging (in a very practical way) with a community of researchers.

We very much appreciate the importance of field work and indeed some of the coauthors are experienced field workers. However, the article in this journal we hope will be read by a diverse audience – from those working on the cognitive neuroscience of perception to those working on the origins of language to those working on specific languages in the field. As such we have been careful to balance the way the paper is written to appeal to a broad audience and therefore have not written it specifically to target linguists working in the field. All that said, we have now added a paragraph in the discussion section that considers the relative merits of different methods and fully acknowledges the trade-offs between the statistically powerful experimental methods we have employed on the one hand and the more ecologically-driven data gathering methods used by some fieldworkers on the other (please see highlighted section at the bottom of page 8).

Regarding the specific number of participants, the nature of calculation of statistical power depends on the number of variables being considered, etc., so it is not possible to provide recommendations regarding specific numbers of consultants/informants that would be required.

R3.4 Recent theoretical linguistic work seems to me to be sympathetic to the paper's point of view. Harbour 2016 and Ackema and Neeleman 2018 are analyses of person systems which take demonstrative systems to be, in essence, at root person systems. In three+ term demonstrative systems, Harbour draws a distinction between those in which the medial term is addressee based and those in which it is distance based. This seems to map onto distinction discussed in lines 302–314.

We thank the reviewer for these suggestions. We now cite the Harbour reference as suggested in the revision (please see the middle highlighted section on page 6).

R3.5 The discussion of two-term systems in the same paragraph is both revealing and, again, a possible point at which to engage with descriptive linguists. Bulgarian is, along with Catalan, a well cited case in which the 'speaker-centred' demonstrative extends into the hearer's space (see Imai 2003 for references; they are central to the works in the last paragraph). I was surprised to see Italian and Mandarin in there too. Maybe I just don't know the descriptive literature well enough but it is possible that the authors have made a discovery here that traditional descriptive grammars have ignored. If so, the discussion should again engage with other researchers, implicitly inviting them to revisit and revise traditional descriptions.

We thank the reviewer for drawing our attention to Imai's (2003) work, which we now cite in the introduction (highlighted sentence on page 3) and in the discussion (highlighted section at the bottom of page 6). We have made it clearer in the results that the addressee effects for the two-term languages are not the same as the effects for the three-term languages (please see the highlighted section in the middle of page 6). For the five three-term languages with addressee effects they are clearly linked to person-centredness, consistent with Harbour. However, for the three two-term languages with addressee effects the results are more consistent with the notion of shared space.

R3.6 Satawal has been described (Yoshida, 1981) as distinguishing in-hand/reachable from near to speaker. The paper implies a reduction of 'near to speaker' to reachable. Is Satawal therefore a potential problem?

We thank the reviewer for raising this point. As Satawal is not one of the languages we include in the sample, and as the evidence base for Satawal is unclear following up the reference, we have chosen not to discuss this point in the manuscript. However, Satawal may well be a language that we hope could be tested in the future using our controlled and powerful method.

R3.7 The notion, expressed in the paper's title and also in line 408, that we conceive of the world in terms of how we can act on it reminded me of Dixon's 1994 study of ergativity. (Again, sorry that I can't provide a page reference as the book is boxed up.) The authors may wish to cite this.

We thank the reviewer for suggesting this reference. Reading Dixon's book, we have chosen not to cite it as we think it is only peripherally relevant.

References

Diessel, H. Demonstratives, joint attention, and the emergence of grammar. *Cognitive Linguistics* **17(4)**, 463-489 (2006).

Diessel, H. & Monakhov, S. Acquisition of demonstratives in cross-linguistic perspective. *Journal of Child Language* (2022). DOI: <https://doi.org/10.1017/S030500092200023X>

Imai, S. Spatial deixis. *How finely do languages divide space?* VDM Verlag (2009).

Kita, S. *Pointing: Where language, cognition and culture meet*. Lawrence Erlbaum (2003).

Küntay, A. & Özyürek, A. Learning to use demonstratives in conversation: what do language specific strategies in Turkish reveal? *J. Child lang.* **33**, 303-320 (2006).

Levinson, S. C. Introduction: Demonstratives - Patterns in diversity. In *Demonstratives in Cross-Linguistic Perspective*, 1st Ed., S. C. Levinson, S. Cutfield, M. Dunn, N. Enfield, S. Meira, Eds. (Cambridge University Press, 2018), pp. 1–35.

Rubio-Fernandez, P. Pragmatic markers: The missing link between language and theory of mind. *Synthese* **199(1)**, 1125-1158 (2020).

Todisco, E., Guijarro-Fuentes, P., Collier, J. & Coventry, K. R. The temporal dynamics of deictic communication. *First Language* **41(2)**, 154-178 (2021).

Decision Letter, first revision:

26th June 2023

Dear Dr. Coventry,

Thank you for your patience as we've prepared the guidelines for final submission of your Nature Human Behaviour manuscript, "Spatial Communication Systems Across Languages Reflect Universal Action Constraints" (NATHUMBEHAV-22092474A). Please carefully follow the step-by-step instructions provided in the attached file, and add a response in each row of the table to indicate the changes that you have made. Please also address the additional marked-up edits we have proposed within the reporting summary. Ensuring that each point is addressed will help to ensure that your revised manuscript can be swiftly handed over to our production team.

We would hope to receive your revised paper, with all of the requested files and forms within two-three weeks. Please get in contact with us if you anticipate delays.

Nature Human Behaviour offers a Transparent Peer Review option for new original research manuscripts submitted after December 1st, 2019. As part of this initiative, we encourage our authors to support increased transparency into the peer review process by agreeing to have the reviewer comments, author rebuttal letters, and editorial decision letters published as a Supplementary item. When you submit your final files please clearly state in your cover letter whether or not you would like to participate in this initiative. Please note that failure to state your preference will result in delays in accepting your manuscript for publication.

In recognition of the time and expertise our reviewers provide to Nature Human Behaviour's editorial process, we would like to formally acknowledge their contribution to the external peer review of your manuscript entitled "Spatial Communication Systems Across Languages Reflect Universal Action Constraints". For those reviewers who give their assent, we will be publishing their names alongside the published article.

Cover suggestions

As you prepare your final files we encourage you to consider whether you have any images or illustrations that may be appropriate for use on the cover of Nature Human Behaviour.

ORCID

Non-corresponding authors do not have to link their ORCIDs but are encouraged to do so. Please note that it will not be possible to add/modify ORCIDs at proof. Thus, please let your co-authors know that if they wish to have their ORCID added to the paper they must follow the procedure described in the following link prior to acceptance:

Nature Human Behaviour has now transitioned to a unified Rights Collection system which will allow our Author Services team to quickly and easily collect the rights and permissions required to publish your work. Approximately 10 days after your paper is formally accepted, you will receive an email in providing you with a link to complete the grant of rights. If your paper is eligible for Open Access, our Author Services team will also be in touch regarding any additional information that may be required to arrange payment for your article. Please note that you will not receive your proofs until the publishing agreement has been received through our system.

Please note that *Nature Human Behaviour* is a Transformative Journal (TJ). Authors may publish their research with us through the traditional subscription access route or make their paper immediately open access through payment of an article-processing charge (APC). Authors will not be required to make a final decision about access to their article until it has been accepted. Find out more about Transformative Journals

[REDACTED]

Best regards,
Alex McKay
Editorial Assistant
Nature Human Behaviour

On behalf of

Jamie

Dr Jamie Horder
Senior Editor
Nature Human Behaviour

Reviewer #1:

Remarks to the Author:

The authors have thoughtfully responded to my main concerns.

There is now some thoughtful discussion about why the distance effect is posited to be discontinuous.

The responses about the potential confounding in the memory game is now clearer and addresses my concern.

The plots are significantly improved, and I think the new Figure 3 is in particular a huge improvement over the old version. I would still recommend the authors think about how they can do analyses and make plots that are even more illuminating. This is such a cool dataset that I still think the plots and analyses could do more to illuminate it. Figure 4 continues to be a sub-optimal way to display the data, in my opinion.

I will reiterate that it is very impressive to have conducted a controlled psycholinguistic study across 29 languages, on a topic of broad interest in the community. I believe it is likely this will be an impactful paper and, in particular, that the data set should be made available so that it can (and will) be used for additional analyses.

Reviewer #2:

Remarks to the Author:

I thank the editor for this opportunity to read the revised resubmission of "Spatial communication

systems across languages reflect universal action constraints". Responses to each point below:

Test and claim of semantic universals:

I appreciate the paper's clearer stance on what it actually tests and claims. This makes it easier to relate to.

Interactional parameters:

I disagree with the authors' stance that addressee location is the key interactional factor. The experiment's different addressee locations in relation to speaker and referent manipulate the spatial layout and perspective – any assumed interactional differences between these variants (and I have trouble understanding the authors' explanation of what these differences amount to) will be merely symptoms of those spatial contrasts. The speaker may have a bit more difficulty gauging the addressee's attention in side-by-side trials than in face-to-face trials, but the protocol does not probe for this, and there is no developed discussion of the issue. It is true that the variants present a contrast in alignment of relevance to perspective-taking but, again, the protocol does not distinguish between spatial and competing variables. The fact remains: from the speaker's/participant's perspective, the addressee's/experimenter's epistemic and attentional relationship to the referents is constant across trials. The current experiment does not test this fundamental interactional dimension and at no point does it step out of its spatial mindset.

Followingly, I fundamentally disagree with the notion that attentional dis/engagement is a tangential aspect of deictic communication. As I'm sure the authors will agree, the purpose of demonstratives is to coordinate attention, and the drawing of attention to a referent not attended to (that is, to attentionally engage the addressee) is arguably one of their most fundamental functions. This holds true whether or not a language has dedicated forms for this purpose (like Turkish), although such languages may offer particular enlightenment on the phenomenon. For the emerging literature on the topic of engagement, uncited by the paper, this is a core tenet (see especially Evans, Bergqvist & San Roque 2017 and works cited therein).

The study unveils a rather remarkable consistency in the mapping of demonstrative distinctions onto the reachable/unreachable contrast across languages. This is interesting but is in itself not evidence of the fundamentality of egocentric spatial encoding. If the paper is serious about advancing this explanation at the expense of "social/interactive" alternatives it should offer a solid methodological paradigm for pitting the explanatory models against each other. In my view it doesn't and, for me, this remains the main weakness of the study.

Participant instructions:

This is still unclear to me. The response to Reviewer 1 and the added explanation in the Procedure description make it clear that participants were (thankfully) not instructed with examples of

demonstratives involving a specific demonstrative at a specific location, and that there was therefore no priming to use demonstratives in a specific way. But instructions did involve priming to use demonstratives as such, and very explicitly so. As I pointed out in my original review, the memory game format may well have helped towards neutralizing participants' awareness that the experiment was about demonstratives, but entering the experiment they must have been fully aware that they were supposed to use this/that-type words. This is at odds with the paper's statement that the "method avoids participants consciously choosing to use demonstratives during the task" (lines 190, 471). The authors should eliminate the confusion here. It would also help to include verbatim examples of the instructions in English and a selection of the other sample languages (glossed and translated into English).

Language sample:

Subtle reference to the limits of the diversity of the sample has been added in the new version. I would have wished a bit more contextualization and problematization of the sample but will not insist on further elaboration.

Cited works:

Evans, N., H. Bergqvist & L. San Roque. 2017. The grammar of engagement I: framework and initial exemplification. *Language & Cognition*. <https://doi.org/10.1017/langcog.2017.21>

Reviewer #3:

Remarks to the Author:

I am satisfied with the authors' responses and look forward to seeing the published paper.

Final Decision Letter:

Dear Professor Coventry,

We are pleased to inform you that your Article "Spatial Communication Systems Across Languages Reflect Universal Action Constraints", has now been accepted for publication in *Nature Human Behaviour*.

Please note that *Nature Human Behaviour* is a Transformative Journal (TJ). Authors whose manuscript was submitted on or after January 1st, 2021, may publish their research with us through the traditional subscription access route or make their paper immediately open access through payment of an

article-processing charge (APC). Authors will not be required to make a final decision about access to their article until it has been accepted. IMPORTANT NOTE: Articles submitted before January 1st, 2021, are not eligible for Open Access publication. Find out more about Transformative Journals

With best regards,

Jamie

Dr Jamie Horder
Senior Editor
Nature Human Behaviour